# *Spinosaurus* is not an aquatic dinosaur

Paul C Sereno[1,2]*, Nathan Myhrvold[3], Donald M Henderson[4], Frank E Fish[5], Daniel Vidal[6], Stephanie L Baumgart[1], Tyler M Keillor[1], Kiersten K Formoso[7,8], Lauren L Conroy[1]

[1]1Department of Organismal Biology, University of Chicago, Chicago, United States; [2]Committee on Evolutionary Biology, University of Chicago, Chicago, United States; [3]Intellectual Ventures, Bellevue, United States; [4]Royal Tyrrell Museum of Palaeontology, Alberta, Canada; [5]Department of Biology, West Chester University, West Chester, United States; [6]Grupo de Biología Evolutiva, UNED, Madrid, Spain; [7]Department of Earth Sciences, University of Southern California, Los Angeles, United States; [8]Dinosaur Institute, Natural History Museum of Los Angeles County, Los Angeles, United States

*For correspondence: dinosaur@uchicago.edu

Competing interest: The authors declare that no competing interests exist.

**Abstract** A predominantly fish-eating diet was envisioned for the sail-backed theropod dinosaur *Spinosaurus aegyptiacus* when its elongate jaws with subconical teeth were unearthed a century ago in Egypt. Recent discovery of the high-spined tail of that skeleton, however, led to a bolder conjecture that *S. aegyptiacus* was the first fully aquatic dinosaur. The 'aquatic hypothesis' posits that *S. aegyptiacus* was a slow quadruped on land but a capable pursuit predator in coastal waters, powered by an expanded tail. We test these functional claims with skeletal and flesh models of *S. aegyptiacus*. We assembled a CT-based skeletal reconstruction based on the fossils, to which we added internal air and muscle to create a posable flesh model. That model shows that on land *S. aegyptiacus* was bipedal and in deep water was an unstable, slow-surface swimmer (<1 m/s) too buoyant to dive. Living reptiles with similar spine-supported sails over trunk and tail are used for display rather than aquatic propulsion, and nearly all extant secondary swimmers have reduced limbs and fleshy tail flukes. New fossils also show that *Spinosaurus* ranged far inland. Two stages are clarified in the evolution of *Spinosaurus*, which is best understood as a semiaquatic bipedal ambush piscivore that frequented the margins of coastal and inland waterways.

## Editor's evaluation

This article evaluates the hypothesis that Spinosaurus was a specialized aquatic dinosaur, by developing a CT-based skeletal restoration and examining its hydrodynamic properties. In this reappraisal of the "aquatic hypothesis", new results support the alternative "semi-aquatic hypothesis". This article will be of interest to vertebrate paleontologists and functional morphologists, as well as wider academic and non-academic audiences.

## Introduction

In 1915, Ernst von Stromer announced the discovery in Egypt's Western Desert of the elongate jaws and partial skeleton of a large sail-backed predator *Spinosaurus aegyptiacus* (**Stromer, 1915**). Other bones found nearby (**Stromer, 1934**) contributed to his initial reconstruction of *S. aegyptiacus* as a sail-backed, piscivorous biped (**Stromer, 1936**), shortly before all of these bones were destroyed in World War II (**Nothdurft et al., 2002**; **Smith et al., 2006**). Over the last 30 years, additional skull and postcranial bones came to light in western Morocco in beds of similar age to those in Egypt (**Russell, 1996**; **Dal Sasso et al., 2005**; **Smyth et al., 2020**; **Ibrahim et al., 2020a**). Central among these

finds was a partial skeleton (designated the neotype) that allowed a more complete reconstruction, confirming its interpretation as a semiaquatic piscivore (*Ibrahim et al., 2014*).

As skeletal information on the unusual predator improved, so has speculation as to whether *S. aegyptiacus* was better adapted to life in water as an aquatic predator, based on inferences from oxygen isotopes in enamel (*Amiot et al., 2010*), the dental rosette likened to the jaws of a conger eel (*Vullo et al., 2016*), the alleged elevated positioning of the orbits in the skull for visibility while largely submerged (*Arden et al., 2019*), the hypothetical underwater role of the trunk sail (*Gimsa et al., 2016*), and the infilling of the medullary cavities of hind limb bones that may have functioned as ballast (*Ibrahim et al., 2014*; *Aureliano et al., 2018*).

## The aquatic hypothesis

Recent discovery of the tall-spined tail bones of the neotypic skeleton reinvigorated the interpretation of *S. aegyptiacus* as the first fully aquatic dinosaur (*Ibrahim et al., 2020b*), here dubbed the 'aquatic hypothesis,' which makes three basic propositions. Unlike any other theropod, according to the hypothesis, *S. aegyptiacus*:

1. reverted to a quadrupedal stance on land, as shown by a trunk-positioned center of mass (*Ibrahim et al., 2014*; *Ibrahim et al., 2020b*), ostensibly knuckle-walking with long-fingered, long-clawed forelimbs;
2. functioned in water as a capable, diving pursuit predator using an expanded tail as a 'novel propulsor organ' (*Ibrahim et al., 2020b*) or as a 'subaqueous forager' (*Fabbri et al., 2022*); and
3. fossils would be found exclusively in coastal or deep-water marine habitats, like all large-bodied secondarily aquatic vertebrates, and would not be expected to be found in freshwater inland environments.

We test these three central propositions.

Critique of the aquatic hypothesis thus far has focused on an alternative functional explanation for the high-spined tail as a display structure and largely qualitative functional interpretations of its skeletal anatomy (*Hone and Holtz, 2021*). Biomechanical evaluation of the aquatic functionality of *S. aegyptiacus* remains rudimentary. The propulsive capacity of the tail in water was judged to be better than terrestrial counterparts by oscillating miniature plastic tail cutouts in water (*Ibrahim et al., 2020b*), a limited approximation of the biomechanical properties of an anguilliform tail (*Lighthill, 1969*; *van Rees et al., 2013*; *Gutarra and Rahman, 2022*) that failed to take account of the bizarre anterior half of the animal. The center of body mass, a critical functional parameter, has been estimated for *S. aegyptiacus* three times, each estimate pointing to a different location ranging from the middle of the trunk (*Ibrahim et al., 2014*; *Ibrahim et al., 2020b*) to a position over the hind limbs (*Henderson, 2018*). Quantitative comparisons have not been made regarding the size or surface area of the limbs, hind feet, and tail of *S. aegyptiacus* to counterparts in extant primary or secondary swimmers.

Thus, adequate evaluation of the aquatic hypothesis requires more realistic biomechanical tests, quantitative body, axial and limb comparisons between *S. aegyptiacus* and extant primary and secondary swimmers, and a survey of bone structure beyond the femur and shaft of a dorsal rib. Such tests and comparisons require an accurate 3D digital flesh model of *S. aegyptiacus*, which, in turn, requires an accurate skeletal model. Hence, we began this study by assembling a complete set of CT scans of the fossil bones for *S. aegyptiacus* and its African forerunner, *Suchomimus tenerensis* (*Sereno et al., 1998*).

## Aquaphilic terminology

Aquatic status is central to the 'aquatic hypothesis.' The hypothesis holds that *S. aegyptiacus* is the first non-avian dinosaur bearing skeletal adaptations devoted to lifestyle and locomotion in water, some of which inhibited terrestrial function. The contention is that *S. aegyptiacus* was not only a diving pursuit predator in the open-water column, but also a quadruped on land with long-clawed forelimbs poorly adapted for weight support. A later publication seemed to downgrade that central claim by suggesting that any vertebrate with 'aquatic habits,' such as wading, submergence, or diving, had an 'aquatic lifestyle' (*Fabbri et al., 2022*). That broadened usage of 'aquatic lifestyle,' however, blurs the long-standing use of aquatic as applied to lifestyle (*Pacini and Harper, 2008*). We outline below the traditional usage of aquaphilic terms, which we follow.

The adjective 'aquatic' is used either as a broad categorization of *lifestyle* or, in more limited capacity, in reference to an *adaptation* of a species or group. In the former case, a vertebrate with an 'aquatic lifestyle' or 'aquatic ecology' is adapted for life primarily, or solely, in water with severely reduced functional capacity on land (*Pacini and Harper, 2008*). *Aquatic vertebrates* (e.g., bony fish, sea turtles, whales) live exclusively or primarily in water and exhibit profound cranial, axial, or appendicular modifications for life in water, especially at larger body sizes (*Webb, 1984*; *Webb and De Buffrénil, 1990*; *Hood, 2020*). For example, extant whales are secondarily aquatic mammals that spend all of their lives at sea and exhibit profound skeletal modifications for aquatic sensory and locomotor function. A marine turtle, similarly, is considered an aquatic reptile, regardless of whether it ventures ashore briefly to lay eggs, because the vast majority of its life is spent in water using profoundly modified limbs for aquatic locomotion (flippers) that function poorly on land.

An aquaphilic animal with less profound adaptations to an aqueous arena is said to be *semiaquatic* (or semi-aquatic), no matter the proportion of aquatic foodstuffs in its diet, the proportion of time spent in water, or the proficiency of swimming or diving. Nearly all semiaquatic vertebrates are secondarily aquaphilic, having acquired aquatic adaptations over time to enhance functional capacity in water without seriously compromising terrestrial function (*Howell, 1930*; *Hood, 2020*). Indeed, semiaquatic animals are also semiterrestrial (*Fish, 2016*). For example, freshwater turtles are regarded as semiaquatic reptiles because they frequent water rather than live exclusively within an aqueous habitat, are sometimes found in inland habitats, and exhibit an array of less profound modifications (e.g., interdigital webbing) for locomotion in water (*Pacini and Harper, 2008*). Likewise, extant crocodylians and many waterbirds are capable swimmers and divers but retain excellent functional capacity on land. Auks (Alcidae), among the most water-adapted of semiaquatic avians, are agile wing-propelled, pursuit divers with an awkward upright posture on land resembling penguins, but they retain the ability to fly and inhabit land for extended periods (*Nettleship, 1996*). On the other hand, the flightless penguins (Sphenisciformes) are considered aquatic due to their more profound skeletal modifications for swimming and deep diving and more limited terrestrial functionality, although still retaining the capacity to trek inland and stand for considerable durations while brooding. As nearly all semiaquatic vertebrates have an aquatic diet and the ability to swim or dive, more profound functional allegiance to water is requisite for an 'aquatic' appellation (*Pacini and Harper, 2008*).

An *aquatic adaptation* of an organism refers to the function of a particular feature, not the overall lifestyle of an organism. That feature should have current utility and primary function in water (*Houssaye and Fish, 2016*). Aquatic adaptations are presumed to have evolved their functionality in response to water and cannot also have special functional utility in a subaerial setting. For example, the downsized, retracted external nares in spinosaurids would inhibit water intake through the nostrils while feeding with the snout submerged (*Sereno et al., 1998*; *Dal Sasso et al., 2005*; *Ibrahim et al., 2014*; *Hone and Holtz, 2021*). There is at present no plausible alternative explanation involving terrestrial function for the downsizing and retraction of the external nares in spinosaurids, a unique condition among non-avian theropods. In contrast, the hypertrophied neural spines of the tail in *S. aegyptiacus* are ambiguous as an 'aquatic adaptation' because expanded tails can function both as aquatic propulsors and terrestrial display structures. For the expanded tail to be an 'aquatic adaptation,' its morphological construction and biomechanical function must unequivocally show primary utility and capability in water, as is the case with extant tail-powered primary or secondarily aquatic vertebrates (e.g., newts, crocodylians, beavers, otters; *Fish et al., 2021*). The same must be shown or inferred to be the case in extinct secondarily aquatic vertebrates (*Gutarra and Rahman, 2022*). Using various comparative and biomechanical approaches (below), we have not found such substantiating evidence to interpret the heightened tail in *S. aegyptiacus* or other spinosaurids as an aquatic adaptation, confirming similar conclusions reached recently by *Hone and Holtz, 2021*.

## Our approach

To test the aquatic hypothesis for *S. aegyptiacus*, we began with CT scans of spinosaurid fossils from sites in Africa to build high-resolution 3D skeletal models of *S. aegyptiacus* (Figure 1A) and its forerunner, *S. tenerensis* (Figure 1F). Many vertebrae and long bones in both genera show significant internal pneumatic (air) or medullary (marrow) space, which has ramifications for buoyancy. When compared to the 2D silhouette drawing used in the aquatic hypothesis (*Ibrahim et al., 2020b*), our CT-based 3D skeletal model of *S. aegyptiacus* differs significantly in skeletal proportions.

We enveloped the skeletal model in flesh informed by CT scans revealing the muscle volume and air spaces in extant reptilian and avian analogs. To create a 3D flesh model for *S. aegyptiacus* (Figure 2A and B), internal air spaces (trachea, lungs, air sacs) were shaped and positioned as in extant analogs. We created three options for internal air volume based on extant squamate, crocodilian, and avian conditionsand assigned densities to body partitions based on local tissue types and air space. We calculated the surface area and volume of the flesh model as well as its component body parts.

We posed this integrated flesh model in bipedal, hybrid- and axial-powered poses, the latter two based on the swimming postures of extant semiaquatic reptiles (*Grigg and Kirshner, 2015*; Figure 2B). We calculated *center of mass* (CM) and *center of buoyancy* (CB) to evaluate the habitual two- or four-legged stance of *S. aegyptiacus* on land (Figure 1A), the depth of water at the point of flotation (Figure 2D), and the neutral position of the flesh model in deeper water (Figure 2A and B). Using biomechanical formulae (*Lighthill, 1969*) and data from extant alligators (*Fish, 1984*), we estimated the maximum force output of its tail, which was used to calculate maximum swimming velocity at the surface and underwater(Figure 3A). We also evaluated its stability, maneuverability, and diving potential in water (Figure 3B), with all of these functional capacities compared to extant large-bodied aquatic vertebrates.

We turned to extant analogs to consider the structure and function of similar spine-supported sails over the trunk and tail in lizards and the form of tail vertebrae in tail-powered secondary swimmers (Figure 4). We also considered the relative size (surface area) of appendages in a range of secondary swimmers (Figure 5), and how the surface area of foot paddles and tail scale in crocodylians (Figure 6).

Lastly, we turned to the spinosaurid fossil record to look at the habitats where spinosaurid fossils have been found. We reviewed their distributionto determine whether spinosaurids, and *S. aegyptiacus* in particular, were restricted to coastal, marine habitats like all large secondarily aquatic vertebrates. We updated spinosaurid phylogeny in order to discern major stages in the evolution of spinosaurid piscivorous adaptations and sail structures (Figure 8), incorporating the latest finds including new fossils of *Spinosaurus* from Niger.

## Institutional abbreviations

BSPG, Bayerische Staatssammlung für Paläontologie und Geologie, Munich, Germany; FMNH, Field Museum of Natural History, Chicago, IL, USA; FSAC, Faculté des Sciences Aïn Chock, University of Casablanca, Casablanca, Morocco; KU, The University of Kansas, Natural History Museum, Lawrence, KS, USA; MNBH, Musée National de Boubou Hama, Niamey, Niger; MNHN, Muséum national d'Histoire naturelle, Paris, France; NMC, Canadian Museum of Nature, Ottawa, Canada; UCMP, University of California, Museum of Paleontology, Berkeley, CA, USA; UCRC, University of Chicago Research Collection, Chicago, IL, USA; UF, University of Florida, University of Florida Collections, Gainesville, FL, USA; UMMZ, University of Michigan, Museum of Zoology, Ann Arbor, MI, USA; WDC, Wildlife Discovery Center, Lake Forest, IL, USA.

## Results

### Spinosaurid skeletal models

Our skeletal reconstruction of an adult *S. aegyptiacus* is just under 14 m long (*Figure 1A*), which is more than 1 m shorter than previously reported (*Ibrahim et al., 2014*). Major differences are apparent when compared to the 2D graphical reconstruction of the aquatic hypothesis (*Ibrahim et al., 2020b*). The length of the presacral column, depth of the ribcage, and length of the forelimb in that reconstruction were overestimated by ~10, 25, and 30%, respectively, over dimensions based on CT-scanned fossils. When translated to a flesh model, all of these proportional overestimates (heavier neck, trunk, forelimb) shift the center of mass anteriorly (see 'Materials and methods').

The hind limb long bones (femur, tibia, fibula, metatarsals) in *S. aegyptiacus* lack the medullary cavity common to most dinosaurs and theropods in particular. When first discovered, the infilled hind limb bones in *S. aegyptiacus* were interpreted as ballast for swimming (*Ibrahim et al., 2014*). However, the infilled condition is variable as shown by the narrow medullary cavity in a femur of another individual slightly larger than the neotype (*Russell, 1996*; NMC 41869). Furthermore, the bone infilling is fibrolamellar and cancellous, similar to the infilled medullary cavities of other large-bodied terrestrial dinosaurs (*Vanderven et al., 2014*) and mammals (*Houssaye et al., 2016*). In contrast, dense pachystotic bone composes the

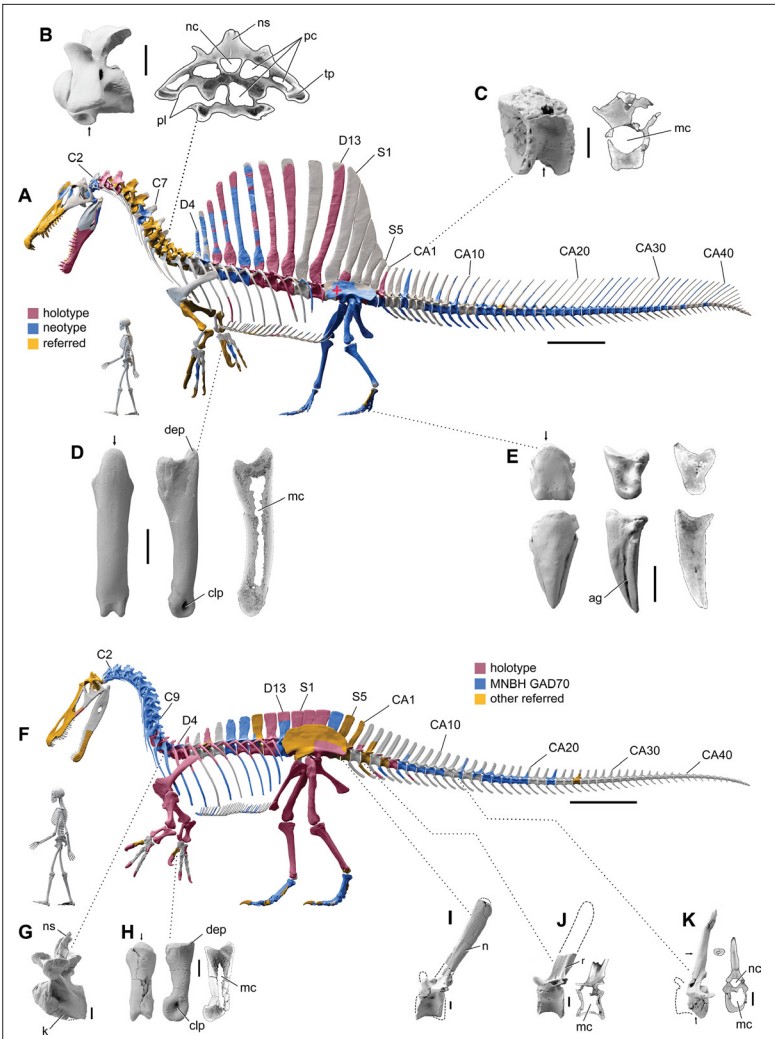

**Figure 1.** Digital skeletal reconstructions of the African spinosaurids *Spinosaurus aegyptiacus* and *Suchomimus tenerensis*. (**A**) *S. aegyptiacus* (early Late Cretaceous, Cenomanian, ca. 95 Ma) showing known bones based on the holotype (BSPG 1912 VIII 19, red), neotype (FSAC-KK 11888, blue), and referred specimens (yellow) and the center of mass (red cross) of the flesh model in bipedal stance (overlap priority: neotype, holotype, referred bones). (**B**) Cervical 9 (BSPG 2011 I 115) in lateral view and coronal cross-section showing internal air space. (**C**) Caudal 1 centrum (FSAC-KK 11888) in anterolateral view and coronal CT cross-section. (**D**) Right manual phalanx I-1 (UCRC PV8) in dorsal, lateral, and sagittal CT cross-sectional views. (**E**) Pedal phalanges IV-4, IV-ungual (FSAC-KK 11888) in dorsal, lateral, and sagittal CT. (**F**) *S. tenerensis* (mid Cretaceous, Aptian-Albian, ca. 110 Ma) showing known bones based on the holotype (MNBH GAD500, red), a partial skeleton (MNBH GAD70, blue), and other referred specimens (yellow) (overlap priority: holotype, MNBH GAD70, referred bones). (**G**) Dorsal 3 in lateral view (MNBH GAD70). (**H**) Left manual phalanx I-1 (MNBH GAD503) in dorsal, lateral, and sagittal CT cross-sectional views. (**I**) Caudal 1 vertebra in lateral view (MNBH GAD71). (**J**) Caudal ~3 vertebra in lateral view (MNBH GAD85). (**K**) Caudal ~13 vertebra in lateral view with CT cross-sections (coronal, horizontal) of the hollow centrum and neural spine (MNBH GAD70). ag, attachment groove; C2, 7, 9, cervical vertebra 2, 7, 9; CA1, 10, 20, 30, 40, caudal vertebra 1, 10, 20, 30, 40; clp, collateral ligament pit; D4, 13, dorsal vertebra 4, 13; dip, dorsal intercondylar process; k, keel; mc, medullary cavity; nc, neural canal; ns, neural spine; pc, pneumatic cavity; pl, pleurocoel; r, ridge; S1, 5, sacral vertebra 1, 5. Dashed lines indicate contour of missing bone, arrows indicate plane of CT-sectional views, and scale bars equal 1 m (**A, F**), 5 cm (**B, C**), 3 cm (**D, E, H–K**) with human skeletons 1.8 m tall (**A, F**).

solid and sometimes swollen bones of many secondarily aquatic vertebrates that use increased skeletal density as ballast (*Houssaye, 2009*).

Medullary space is present in most forelimb bones in both *S. aegyptiacus* and *S. tenerensis* (*Figure 1D and H*). The centra of anterior caudal vertebrae are occupied by a large medullary space

(*Figure 1C and J*), and large air-filled pneumatic spaces are present in the centra and neural arches of cervical vertebrae (*Evers et al., 2015*; *Figure 1B*). Collectively, these less dense, internal marrow- and air-filled spaces in *S. aegyptiacus* more than offset the added density of infilled medullary space in the relatively reduced hind limb long bones (*Figure 1A*). Hind limb bone infilling is better explained as compensation for the reduced size of the hind limb long bones that must support a body mass at the upper end of the range for theropods. Bending strength increases by as much as 35% when the medullary cavity is infilled (see Appendix 1).

## *S. aegyptiacus* flesh model form and function

We added flesh to the adult skeletal model and divided the flesh model into body partitions adjusted for density. Muscle volume was guided by CT cross-sections from extant lizards, crocodylians, and birds (*Figure 2B*), and internal air space (pharynx-trachea, lungs, paraxial air sacs) was modeled on lizard, crocodilian, and avian conditions (*Figure 2C*; see 'Materials and methods,' Appendix 2). Whole-body and body part surface area and volume were calculated, and body partitions were assigned density comparable to that in extant analogs (see 'Materials and methods'). For biomechanical analysis, we positioned the integrated flesh model in bipedal stance (*Figure 1A*) as well as hybrid- and axial-powered swimming poses (*Grigg and Kirshner, 2015*; *Figure 2A and B*).

The CM and CB of the flesh model were determined to evaluate habitual stance on land and in shallow water (*Figure 1A*), the water depth at the point of flotation (*Figure 2D*), and its swimming velocity, stability, maneuverability, and diving potential in deeper water (*Figure 3*). No matter the included volume of internal air space, CM is positioned over the ground contact of symmetrically positioned hind feet (*Figure 1A*, red cross). Thus, *S. aegyptiacus* had a bipedal stance on land as previously suggested (*Henderson, 2018*), contrary to trunk-centered CM of the aquatic hypothesis (*Ibrahim et al., 2020b*). Consistent with a bipedal stance, the manus is adapted for prey capture and manipulation (elongate hollow phalanges, scythe-shaped unguals) rather than weight support (*Figure 1A and D*).

Adult *S. aegyptiacus* can feed while standing in water with flotation occurring in water deeper than ~2.6 m (*Figure 2D*). In hybrid or axial swimming poses, trunk air space tilts the anterior end of the model upward (*Figure 2A and B*). With density-adjusted body partitions and avian-like internal air space, the flesh model of *S. aegyptiacus* has a body mass of 7390 kg and an average density of 833 kg/m$^3$ (see 'Materials and methods'), which is considerably less than the density of freshwater (1000 kg/m$^3$) and saltwater (1026 kg/m$^3$) or the average density of living crocodylians (1080 kg/m$^3$; *Grigg and Kirshner, 2015*).

*Swimming velocity* at the surface and underwater in extant lizards and crocodylians is powered by foot paddling and axial undulation (hybrid swimming; *Frey and Salisbury, 2001*) and at moderate to maximum (critical) speeds by axial undulation alone (axial swimming) (*Fish, 1984*; *Grigg and Kirshner, 2015*). We used Lighthill's bulk momentum formula to estimate maximum surface and underwater swimming velocity for the flesh model of *S. aegyptiacus* (*Lighthill, 1969*). Assuming a fully compliant *Alligator*-like tail (tail amplitude 0.24/body length, tail wavelength 0.57/body length, and tailbeat frequency 0.25 Hz; *Fish, 1984*; *Sato et al., 2007*), tail thrust ($P_t$) and maximum velocity (U) can be determined ($P_t = -164.93 + 1899.1U - 896.35U^2$). Assuming turbulent conditions, a body drag coefficient of 0.0035 was estimated for a Reynolds number of 752,400 at a swimming speed of 1.0 m/s. The total power from estimates of drag increased three- to fivefold to account for undulation of the tail, near-surface wave formation, and increased sail drag when underwater (*Figure 3A*). The addition of the sail increases the drag on the body of *S. aegyptiacus* by 33.4%. The intersection of the thrust power curve and drag power curves, where the animal would be swimming at a constant velocity, indicates slow maximum velocity at the surface (~0.8 m/s) and only slightly greater when submerged (~1.4 m/s) (*Figure 3A*). Maximum tail thrust in *S. aegyptiacus* is 820 Watts (683 N or 154 lbs), a relatively low value for the considerable caudal muscle mass in this large theropod (*Snively and Russell, 2007*; *Mallison et al., 2015*). Only a minor amount of caudal muscle power, however, is imparted to the water as thrust during undulation. As a result, maximum velocity is only 1.2 m/s, an order of magnitude less than extant large-bodied (>1 m) pursuit predators. These species (mackerel sharks, billfish, dolphins, and killer whales) are capable of maximum velocities of 10–33 m/s (*Tinsley, 1984*; *Fish, 1998*; *Fish and Rohr, 1999*; *Iosilevskii and Weihs, 2008*).

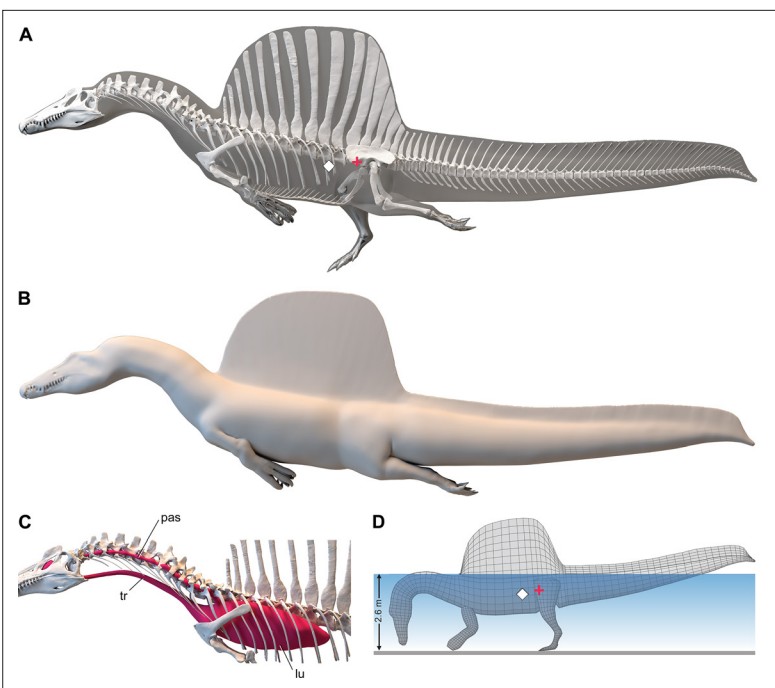

**Figure 2.** Digital flesh model of *Spinosaurus aegyptiacus*. (**A**) Translucent flesh model in hybrid swimming pose showing centers of mass (red cross) and buoyancy (white diamond). (**B**) Opaque flesh model in axial swimming pose with adducted limbs. (**C**) Modeled air spaces ('medium' option) include pharynx-trachea, lungs and paraxial air sacs. (**D**) Wading-strike pose at the point of flotation (2.6 m water depth) showing center of mass (red cross) and buoyancy (white diamond). lu, lungs; pas, paraxial air sacs; tr, trachea.

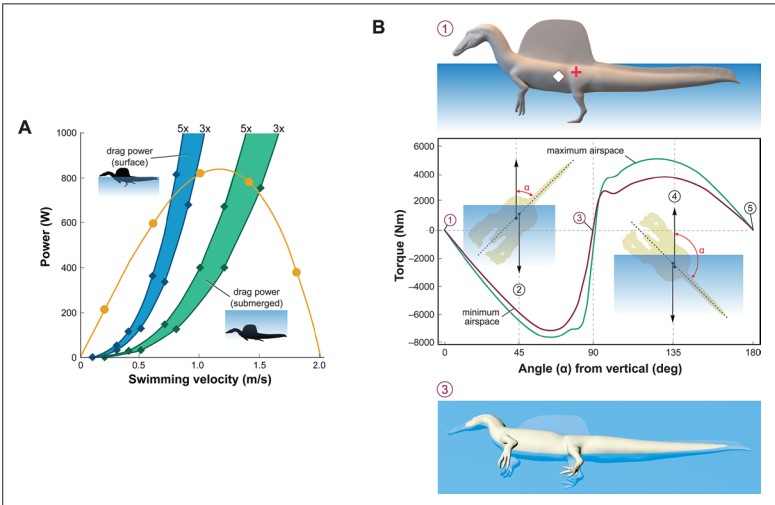

**Figure 3.** Biomechanical evaluation of *Spinosaurus aegyptiacus* in water. (**A**) Tail thrust (yellow curve) and opposing drag forces as a function of swimming velocity at the surface (blue) and submerged (green), with drag during undulation estimated at three and five times stationary drag. (**B**) Stability curve for the flesh model of *S. aegyptiacus* in water showing torque between the centers of mass (red cross) and buoyancy (white diamond), unstable equilibria when upright or upside down (positions 1, 5), and a stable equilibrium on its side (position 3) irrespective of the volume of internal air space. Curves are shown for flesh models with minimum (magenta) and maximum (green) air spaces with a dashed line showing the vertical body axis and vector arrows for buoyancy (up) and center of mass (down).

*Stability* and the capacity to right are important in water. When positioned upright in water, the trunk sail of *S. aegyptiacus* is emergent (*Figure 3B*, position 1). The flesh model, however, is particularly susceptible to long-axis rotation given the proximity of CM and CB, with stable equilibrium attained when floating on its side (*Figure 3B*, position 3). Righting requires substantial torque (~5000 Nm) that is impossible to generate with vertical limbs and a tail with far less maximum force output (~700 N). This stability predicament remains even with the smallest internal air space. The absence of vertical stability and righting potential in water stands in stark contrast to the condition in extant crocodylians and marine mammals (*Fish, 1998*; *Grigg and Kirshner, 2015*).

*Maneuverability* in water (acceleration, turning radius, and speed) wanes as body length increases (*Domenici, 2001*; *Parson et al., 2011*; *Domenici et al., 2014*; *Hirt et al., 2017*; *Gutarra and Rahman, 2022*), which is further compromised in *S. aegyptiacus* by its rigid trunk (see below) and expansive, unretractable sail. In contrast, large-bodied secondary swimmers capable of pursuit predation in open water have fusiform body forms with a narrow caudal peduncle for efficient tail propulsion (ichthyosaurs, cetaceans; *Motani, 2009*), control surfaces for reorientation, and narrow extensions (bills) to enhance velocity in close encounters with smaller more maneuverable prey (*Maresh et al., 2004*; *Domenici et al., 2014*). Besides some waterbirds, semiaquatic pursuit predators are rare and include only the small-bodied (<2 m), exceptionally maneuverable otters that employ undulatory swimming (*Fish, 1994*).

*Diving* with an incompressible trunk requires a propulsive force ($F_g$) greater than buoyancy. For *S. aegyptiacus*, in addition, a depth of ~10 m is needed to avoid wave drag (*Figure 3A*, bottom). The propulsive force required to dive is ~17,000 N: $\left( V_{body} \times \left[ \rho_{Saltwater} - \bar{\rho}_{FleshModel} \right] \times g; 8.94 \text{ m}^3 [1026–833 \text{ kg/m}^3] 9.8 \text{ m/s} = 16,909 \text{ N} \right)$, or ~25 times the maximum force output of the tail. Even with lizard-like internal air space, diving still requires ~15 times maximum force output of the tail. To initiate a dive, furthermore, the tail would be lifted into the air as the body rotates about CB (*Figure 2D*), significantly reducing tail thrust. The now common depictions of *S. aegyptiacus* as a diving underwater pursuit predator contradict a range of physical parameters and calculations, which collectively characterize this dinosaur as a slow, unstable, and awkward surface swimmer incapable of submergence.

## Axial comparisons to aquatic vertebrates and sail-backed reptiles

Axial flexibility is requisite for axial-propulsion in primary or secondary swimmers. However, in *S. aegyptiacus*, trunk and sacral vertebrae are immobilized by interlocking articulations (hyposphene-hypantrum), an expansive rigid dorsal sail composed of closely spaced neural spines, and fused sacral centra (*Figure 1A*).

The caudal neural spines in *S. aegyptiacus* stiffen a bone-supported tail sail by an echelon of neural spines that cross several vertebral segments, which effectively resist bending at vertebral joints (*Figure 4A*). The caudal centra in *S. aegyptiacus* have nearly uniform subquadrate proportions along the majority of the tail in lateral view, rather than narrowing, spool-shaped centra in crocodylians and other secondarily aquatic squamates (*Figure 4D*), which increases distal flexibility during tail undulation. These salient structural features of the tail suggest that it functioned more as a pliant billboard than flexible fluke.

No primary or secondary vertebrate swimmer has a comparable drag-magnifying, rigid dorsal sail including sailfish, the dorsal fin of which is fully retractable and composed of pliable spines in membrane (*Domenici et al., 2014*). In contrast, the distal tail of secondary swimmers such as crocodylians (*Grigg and Kirshner, 2015*), mosasaurs (*Lindgren et al., 2013*) and cetaceans (*Fish, 1998*) is expanded with pliable soft tissues free of bone to form a flexible caudal paddle or fluke (*Figure 4C*).

Spine-supported, torso-to-caudal sails aligned with median cranial crests, in contrast, have evolved multiple times for intraspecific display rather than aquatic propulsion among extant lizard (agamids, iguanians, chameleons). Semiaquatic sailfin and basilisk lizards (*Figure 4B*), for example, do not use their sails while swimming, spend very little time submerged, and are not aquatic pursuit predators (*Hone and Holtz, 2021*).

Caudal centra proportions in most secondary swimmers, as mentioned above, grade from subquadrate to spool-shaped in the distal half of the tail to increase flexibility and undulatory amplitude (*Figure 4D*), whereas those in *S. aegyptiacus* maintain relatively uniform proportions along the tail. This uniformity of subquadrate proportions in *S. aegyptiacus* should not be confused with a more

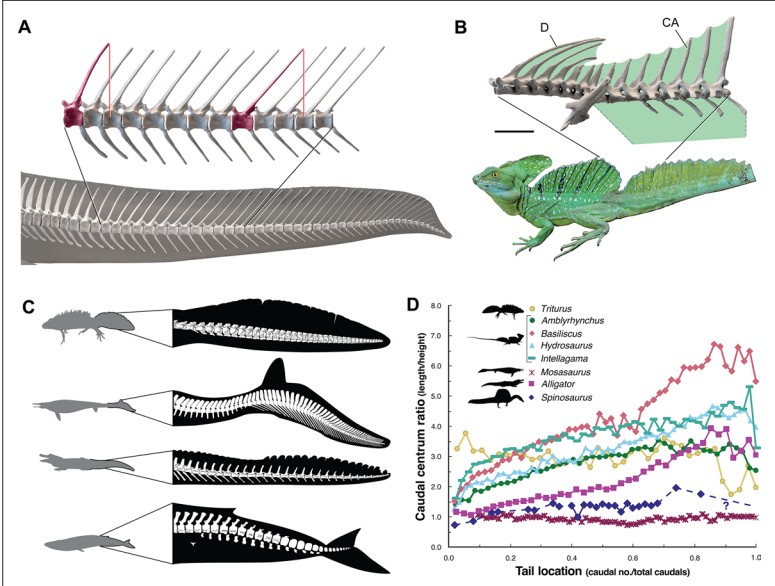

**Figure 4.** Skeletal comparisons between *Spinosaurus aegyptiacus*, a basilisk lizard and secondarily aquatic vertebrates. (**A**) Tail in *S. aegyptiacus* showing overlap of individual neural spines (red) with more posterior vertebral segments. (**B**) Sail structure in the green basilisk (CT-scan enlargement) and in vivo form and coloration of the median head crest and sail (*Basiliscus plumifrons* FMNH 112993). (**C**) Structure of the tail fluke in a urodele, mosasaur, crocodilian, and whale. (**D**) Centrum proportions along the tail in the northern crested newt (*Triturus cristatus* FMNH 48926), semiaquatic lizards (marine iguana *Amblyrhynchus cristatus* UF 41558, common basilisk *Basiliscus basiliscus* UMMZ 121461, Australian water dragon *Intellagama lesueurii* FMNH 57512, sailfin lizard *Hydrosaurus amboinensis* KU 314941), an extinct mosasaurid (*Mosasaurus* sp. UCMP 61221; *Lindgren et al., 2013*), an alligator (*Alligator mississippiensis* UF 21461), and *Spinosaurus* (*S. aegyptiacus* FSAC-KK 11888). Data in Appendix 2.

derived piscine pattern of uniform, short, disc-shaped centra that has evolved in parallel in mosasaurs (*Lindgren et al., 2013*; *Figure 4D*, Appendix 2).

## Appendicular comparisons to vertebrate secondary swimmers

Appendage (fore and hind limb) surface area in secondary swimmers is minimized to reduce drag because terrestrial limbs are inefficient aquatic propulsors. Appendage surface area in *S. aegyptiacus*, in contrast, is substantially greater than in reptilian and mammalian secondary swimmers and even exceeds that of the terrestrial predators *Allosaurus* and *Tyrannosaurus* (*Figure 5*).

Interdigital webbing is used by some secondary swimmers to increase the area of the foot paddle (*Fish, 2004*). Extant crocodylians use their limbs in paddling only at launch and slow speed before tucking them against the body (*Grigg and Kirshner, 2015*). Crocodylian interdigital webbing, which is better developed and always present in the hind foot (*Figure 6C and D*), only modestly increases surface area (<20%). Across a range of body size, we show that crocodylian paddle area scales isometrically (*Figure 6F*; see Appendix 3). The crocodylian foot paddle, thus, becomes even less effective as a propulsor with increasing body size. A crocodylian of spinosaurid size, nonetheless, would have a foot paddle area an order of magnitude greater than is possible in *S. aegyptiacus* (*Figure 6E*). Even a fully webbed hind foot in *S. aegyptiacus* (*Figure 6A*), for which there is no hard evidence to establish as likely, is far too small to have functioned either for significant aquatic propulsion or for stabilizing control.

## Paleohabitats and evolution

Most *Spinosaurus* fossils come from marginal basins along northern Africa in deltaic sediment laid down during an early Late Cretaceous transgression (*Figure 7A*, sites 1, 2). These deposits, however, also include the majority of non-spinosaurid dinosaur remains, all of which may have been transported to some degree from inland habitats to coastal delta deposits. Because fossil transport is one way

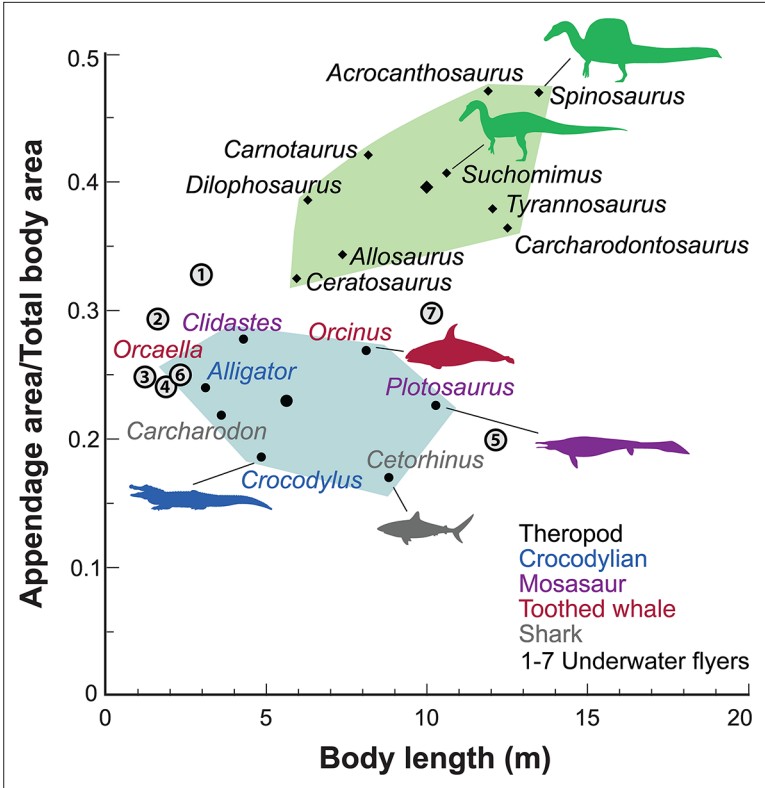

**Figure 5.** Appendage versus total body surface area in aquatic and semiaquatic vertebrates. *Spinosaurus aegyptiacus* and other non-avian theropods (green polygon, centroid large diamond) have appendages with considerable surface area compared to aquatic and semiaquatic vertebrates (blue polygon, centroid large dot). Underwater fliers (1–7 circled), which propel themselves with lift-based wings, also have less overall appendage surface area than in *S. aegyptiacus* and other non-avian theropods. Underwater fliers: 1, plesiosaur *Cryptoclidus oxoniensis*; 2, leatherback sea turtle *Dermochelys coriacea*; 3, emperor penguin *Aptenodytes forsteri*; 4, sea lion *Zalophus californianus*; 5, elasmosaur *Albertonectes vanderveldei*; 6, nothosaur *Ceresiosaurus calcagnii*; 7, pliosaur *Liopleurodon ferox*.

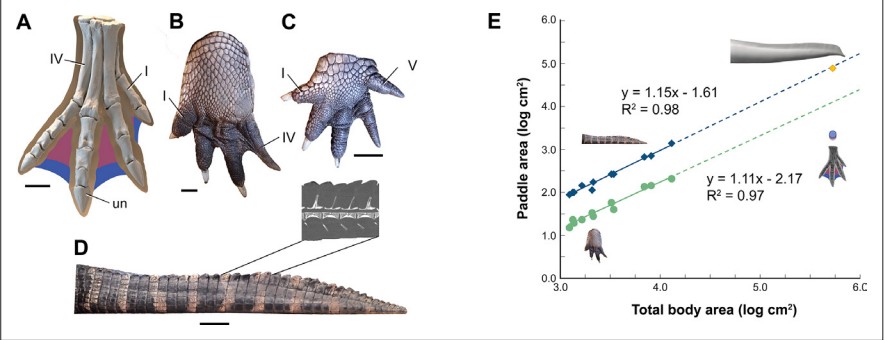

**Figure 6.** Appendage surface area and scaling of paddle surface areas in crocodylians compared to *S. aegyptiacus*. (**A**) Right hind foot of *Spinosaurus aegyptiacus* (FSAC-KK 11888) showing the outlines of digital flesh based on the living ostrich (*Struthio camelus*) as well as partial (pink) and full (blue) interdigital webbing. (**B**) Hind foot of an adult *Alligator mississippiensis* (WDC) in ventral view. (**C**) Forefoot of an adult *A. mississippiensis* (WDC) in ventral view. (**D**) Tail of an adult *A. mississippiensis* (WDC) in lateral view with CT visualization of vertebrae within the fleshy tail fluke. (**E**) Log-log plot of surface areas of webbed hind foot and side of the tail as a function of total body area in a growth series for *A. mississippiensis* (hind foot, green dots; tail, blue diamonds) and adult *S. aegyptiacus* (hind foot, purple-blue dots; tail, yellow diamond). I, IV, V, digits I, IV, V; un, ungual. Scale bars are 10 cm (**A**) and 3 cm (**B–D**).

(downstream), documenting the inland fossil record is key to understanding true habitat range. We recently discovered fossils pertaining to *Spinosaurus* in two inland basins in Niger far from a marine coastline (*Figure 7A*, site 3). They were buried in fluvial overbank deposits alongside terrestrial herbivores (rebbachisaurid and titanosaurian sauropods) (see Appendix 4).

The inland location of these fossils completely undermines the interpretation of *S. aegyptiacus* as a 'highly specialized aquatic predator that pursued and caught its prey in the water column' (*Ibrahim et al., 2020b*). All large-bodied secondarily aquatic vertebrates are marine—both extant (e.g., sea turtles, sirenians, seals, whales) and extinct (e.g., protostegid turtles, ichthyosaurs, metriorhynchoid crocodylomorphs, plesiosaurs). None of these diverse water dwellers live in both saltwater and freshwater habitats (*Evers et al., 2019*; *Motani and Vermeij, 2021*). Secondarily aquatic vertebrates that live in freshwater habitats have marine antecedents and are all small-bodied, such as river dolphins (<2.5 m length; *Hamilton et al., 2001*), small lake-bound seals (<2 m; *Fulton and Strobeck, 2010*), the river-bound Amazonian manatee (<2.5 m; *Guterres-Pazin, 2014*), and a few mosasaurs and plesiosaurs of modest body size (*Gao et al., 2019*).

Large-bodied *semiaquatic* reptiles, in contrast, frequent coastal and inland locales today and in the past. *Sarcosuchus imperator*, among the largest of semiaquatic reptiles (~12 m length; *Sereno et al., 2001*), lived in the same inland basin as *S. tenerensis*. The fossil record supports our interpretation of *Spinosaurus* as a semiaquatic bipedal ambush predator that frequented the margins of both coastal and inland waterways.

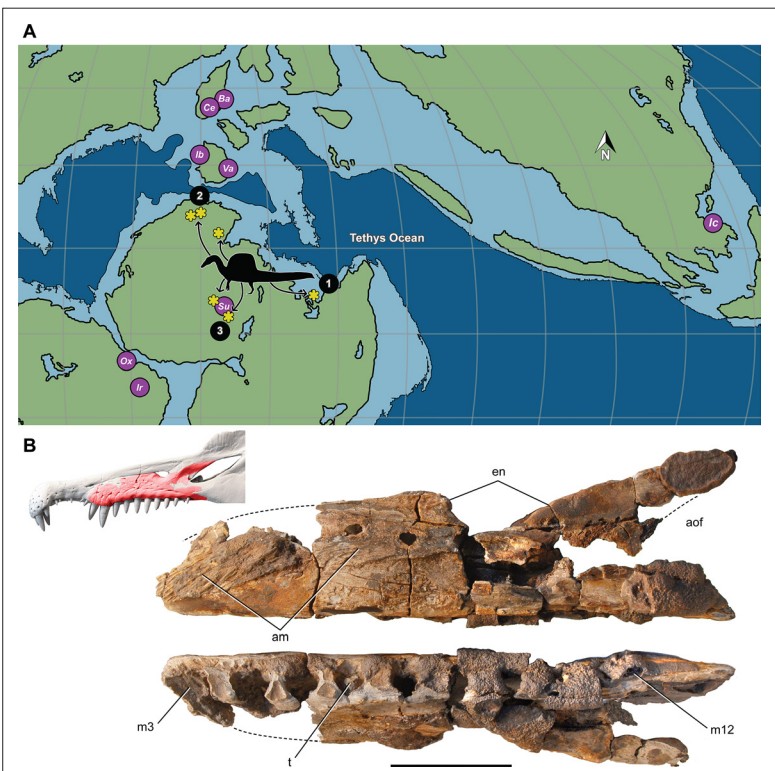

**Figure 7.** Paleogeographic location of spinosaurid fossils. (**A**) Paleogeographic map (early Albian, ~110 Mya; *Scotese, 2014*). showing the circum-Tethyan fossil localities for baryonychines (*Baryonyx, Suchomimus*) and spinosaurines (*Ichthyovenator, Vallibonavenatrix, Oxalaia, Irritator/Angaturama, Spinosaurus*). *Spinosaurus* localities (yellow asterisks) range across northern Africa from coastal (sites 1, 2) to inland (site 3) sites. (**B**) *Spinosaurus* sp. right maxilla (MNBH EGA1) from Égaro North (central Niger) in medial (top) and ventral (bottom) views and shown (red) superposed on the snout of *Spinosaurus aegyptiacus*. 1, *S. aegyptiacus* holotype (Bahariya, Egypt); 2, *S. aegyptiacus* neotype (Zrigat, Morocco); 3, *Spinosaurus* sp. (Égaro North, Niger); am, articular rugosities for opposing maxilla; aofe, antorbital fenestra; Ba, *Baryonyx walkeri*; en, external naris; Ic, *Ichthyovenator laosensis*; Ir, *Irritator challengeri/Angaturama limai*; m3, 12, maxillary alveolus 3, 12; Ox, *Oxalaia quilombensis*; Su, *Suchomimus tenerensis*; t, tooth; Va, *Vallibonavenatrix cani*. Scale bar is 10 cm.

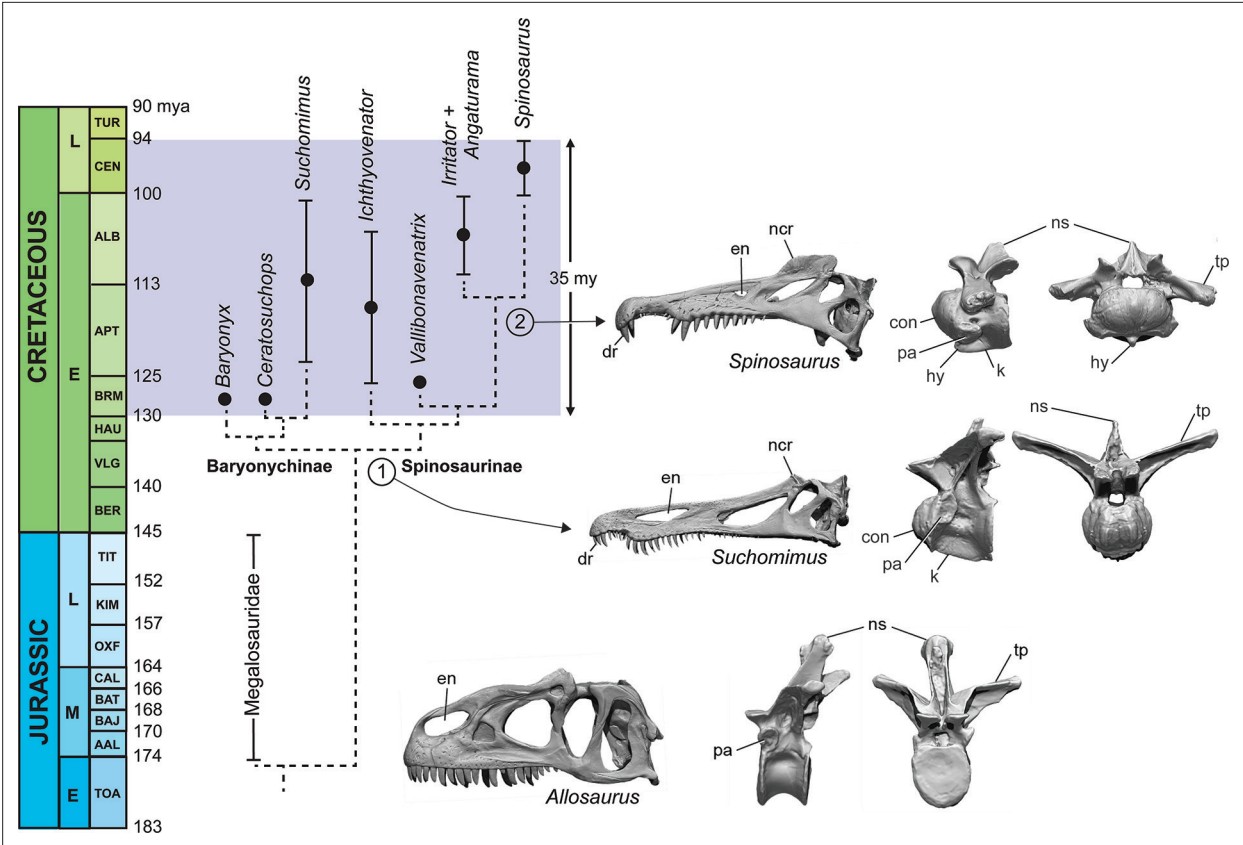

**Figure 8.** Calibrated phylogeny of spinosaurids (Barremian to Cenomanian, ~35 My). Updated phylogenetic analysis of spinosaurids resolves two stages in the evolution of piscivory and display. We show key cranial adaptations in the skull and highlight changes at the anterior end of the trunk to enhance neck ventroflexion (second dorsal vertebra in lateral and anterior views). Bottom, the fully terrestrial theropod *Allosaurus fragilis* (***Madsen, 1976***); middle, the baryonychine spinosaurid *Suchomimus tenerensis* (MNBH GAD70); top, the spinosaurine *Spinosaurus aegyptiacus* (BSPG 1912 VIII 19). con, condyle; dr, dental rosette; en, external naris; hy, hypopophysis; k, keel; ncr, nasal crest; ns, neural spine; pa, parapophysis; tp, transverse process.

The large body size of *S. aegyptiacus* and older related species such as *S. tenerensis* also mitigates against an aquatic interpretation for the former as it would constitute the only instance among vertebrates where the evolution of a secondarily aquatic species occurred at body size greater than 2–3 m. The profound changes involved in fully reentering the aquatic realm from a land-based lifestyle seem more likely to occur at relatively small body size. All other large-bodied secondarily aquatic vertebrates (e.g., ichthyosaurs, plesiosaurs, metriorhynchoid crocodylomorphs, protostegid turtles, mosasaurs, sirenians, whales) evolved the adaptations requisite for an aquatic lifestyle at small body size, increasing in body size once fully established within the marine realm (***Domning, 2000***; ***Thewissen et al., 2009***; ***Polcyn et al., 2014***; ***Moon and Stubbs, 2020***; ***Motani and Vermeij, 2021***).

Phylogenetic analysis of an enlarged dataset for spinosaurids clarifies piscivorous adaptations in the earliest spinosaurids (stage 1, ~130 Ma) that enhance prey capture in shallow water and heighten visual display (***Figure 8***; Appendix 5). In the skull, these include an elongate snout tipped with a dental rosette for snaring fish, retracted external nares to inhibit water intake, and a prominent nasal crest (Charig and Milner, 1997; ***Sereno et al., 1998***). The ornamental crest over the snout is accompanied by the evolution of a postcranial sail of varying height supported by neural spines of the posterior dorsal, sacral and caudal vertebrae (***Stromer, 1915***; ***Sereno et al., 1998***; ***Allain et al., 2012***; ***Barker et al., 2021***).

The earliest spinosaurids, in addition, have 'cervicalized' anterior trunk vertebrae to enhance ventroflexion and the effective length of the neck, presumably as an adaptation to feeding in water (***Hone and Holtz, 2021***). Using the second dorsal vertebrae of the terrestrial predator *Allosaurus* for comparison, the homologous vertebra in spinosaurids shows marked modification (anterior face is convex, prominent ventral keel for muscular attachment, neural spine is reduced, zygapophyses large

and planar). Giraffids, for a similar purpose, have 'cervicalized' the first thoracic vertebra to facilitate dorsiflexion and effective neck length (*Lankester, 1908*; *Danowitz et al., 2015*; *Müller et al., 2021*). Neural spines over the trunk and tail are heightened to varying degrees in all spinosaurids including baryonychines (*Figure 1F and I*). The features cited above to enhance piscivory and display appear to be shared by all currently known spinosaurids and comprise the adaptations we identify as Stage 1 (*Figure 8*).

Baryonychines (e.g., *Suchomimus*, *Baryonyx*, *Ceratosuchops*) have a low, cruciate nasal crest and swollen brow ridges over the orbits for display or agonistic purposes. These may comprise features unique to this subclade of spinosaurids. Spinosaurines, on the other hand, exhibit further specializations for piscivory and display (*Figures 1A and 8*, stage 2). Piscivorous adaptations include spaced teeth with smooth carinae for puncturing efficiency, smaller, more retracted external nares to inhibit water intake, more prominent muscle attachments on the ventral aspect of cervicodorsal vertebrae for ventral lunging, and scythe-shaped manual unguals for slicing (*Sues et al., 2002*; *Dal Sasso et al., 2005*; *Ibrahim et al., 2014*). Adaptations for enhanced display include a heightened cranial crest, low cervical sail, and a hypertrophied torso-to-caudal sail.

## Discussion

In 1915 Ernst Stromer highlighted the remarkable adaptations in the jaws and neural spines of *S. aegyptiacus* for piscivory and ostentatious display, respectively, citing modern analogs for both (*Stromer, 1915*). Nothing close to this morphology had ever been described among nonavian dinosaurs at that time. More recently, a sequence of investigators have gone further, attempting to fathom the manner in which the lifestyle of this large predatory dinosaur engaged coastal waters. All have been hamstrung by the scarcity and fragmentary nature of the specimens, as all of Stromer's Egyptian fossils were destroyed in World War II. Indeed, the unveiling of a new partial skeleton from Morocco (*Ibrahim et al., 2014*) and its tail 6 years later (*Ibrahim et al., 2020b*) generated hypotheses for semi-aquatic and aquatic interpretations, respectively.

The superficially eel-like morphology of the tail, viewed as a 'novel propulsor organ,' provided the inspiration for the 'aquatic hypothesis,' which envisioned *S. aegyptiacus* as a tail-propelled, diving predator 'that pursued and caught its prey in the water column' (*Ibrahim et al., 2020b*). Conversely, as would be requisite for status as a secondarily aquatic reptile, its terrestrial capabilities were regarded as seriously diminished by a trunk-positioned center of body mass (*Ibrahim et al., 2014*; *Ibrahim et al., 2020b*) that would require a quadrupedal stance on land and the use of long-clawed forelimbs not at all designed for weight support. Presented as support for the aquatic hypothesis, Fabbri et al used bone compactness to assert that *S. aegyptiacus* was a 'subaqueous forager' with diving bona fides (*Fabbri et al., 2022*).

The aquatic hypothesis, nonetheless, requires far more than proving its tail was a high-powered source of propulsion or its bones a bit more compact. In order to conclude that *S. aegyptiacus* was an aquatic diver and pursuit predator, one also must understand its buoyancy, stability, velocity, maneuverability, and diving performance in water. Those calculations require an accurate flesh rendering, which in turn is built over an accurate skeletal model.

Therefore, we began with CT scans of the fossils to piece together an accurate skeletal model, discovering major discrepancies with the original 3D skeletal model (*Ibrahim et al., 2014*) and the 2D skeletal silhouette used by the aquatic hypothesis (*Fabbri et al., 2022*). Comparisons to the 2D model with the more accurate tail show that skeletal regions anterior to the hips are enlarged in length and depth beyond the dimensions of our CT-based reconstruction, shifting the CM in the resulting flesh model forward from the hips to the trunk. Trunk length was increased in both previous models of *S. aegyptiacus* due to unnatural ventroflexion of the dorsal column that also spread further the neural spines of the sail (*Figure 9*). When neotype (CT-scanned) or rebuilt holotype dorsal vertebrae of *S. aegyptiacus* are rearticulatd in an osteological neutral pose, the shorter torso has a straighter column with less spread neural spines. The ribcage, in addition, is not as deep, based on the preserved rib pieces of the holotype and neotype and the nearly complete ribcage known for *S. tenerensis* (*Figures 1F and 9*). These proportions effectively reduce the volume of the trunk in our flesh model (*Figure 2*). The flesh model used by the aquatic hypothesis, likewise, underestimated the muscle mass at the base of the tail, judging from our study of CT scans of crocodylians and a range of other reptiles (*Díez Díaz et al., 2020*). These differences are far from trivial when considering centers of

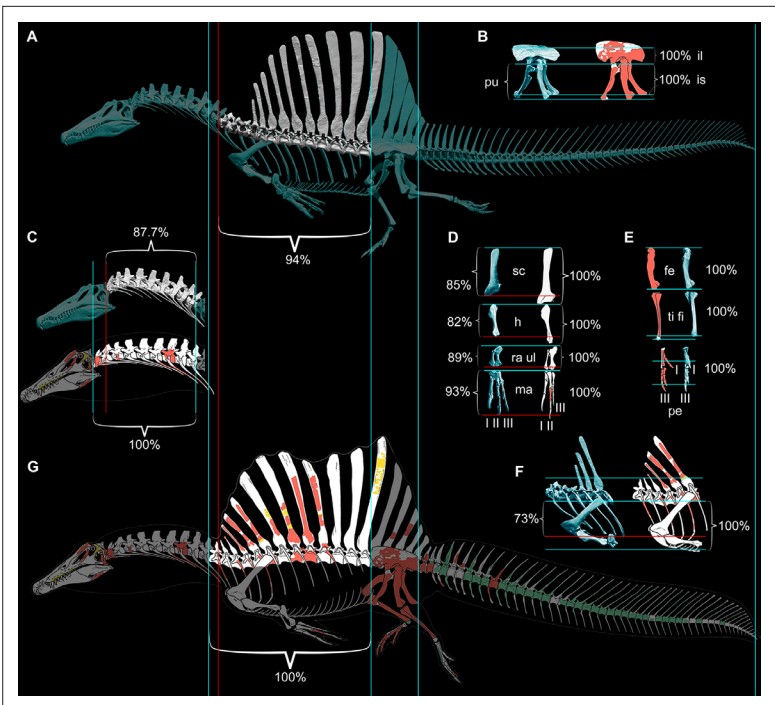

**Figure 9.** Comparison of skeletal reconstructions for *Spinosaurus aegyptiacus* in left lateral view. (**A**) Digital skeletal reconstruction from this study in left lateral view. (**B**) Pelvic girdle. (**C**) Cervical column (C1–10). (**D**) Pectoral girdle and forelimb. (**E**) Hind limb. (**F**) Anterior trunk. (**G**) Silhouette skeletal drawing from the aquatic hypothesis (from *Ibrahim et al., 2020b*). On one side of each length comparison, one or two blue lines are shown that register the alternative reconstructions. The opposing end of each length comparison either has a single blue line (when comparisons match, both 100%) or a red line as well for the shorter one (<100%): A blue line on the right or top sides of each comparison is used for registration. The opposing side has a blue line if reconstructions agree on length (100%), or a blue line for the length estimate in this study and a red line for that of the aquatic hypothesis. In all disparate comparisons, the reconstruction in this study is shorter (percentage given). Skeletal reconstructions (**A, G**) are aligned by the anterior and posterior margins of the ilium and measured to the cervicodorsal junction (C10-D1); the pelvic girdle (**B**) is aligned along the ventral edge of the sacral centra and base of the neural spines and measured to the distal ends of the pubis and ischium; the cervical column (**C**) is aligned at the cervicodorsal junction (C10-D1) and measured to the anterior end of the axis (C2); the scapula and components of the forelimb (humerus, ulna, manual digit II, manual phalanx II-1) (**D**) are aligned at the distal end of the blade and their proximal ends, respectively, and measured to the opposing end of the bone; the components of the hind limb (femur, tibia, pedal digits I, III) (**E**) are aligned at their proximal ends and measured to the opposing end of the bone; and anterior trunk depth (**F**) is aligned along the ventral edge of the centrum and neck of the spine of D6 and measured to the ventral edge of the coracoid. II-1, manual phalanx II-1; Ili, ilium; F, femur; H, humerus; Ish, ischium; l, left; Pe, pes; Pu, pubis; Ma, manus; r, right; RaU, radius-ulna; Sc, scapula; TF, tibia-fibula.

---

mass and buoyancy in *S. aegyptiacus*. The dinosaur, in fact, stood back up on its hind legs like all other theropods.

With a more accurate flesh model in hand, we embarked on a range of biomechanical tests of its performance in water, determining that it fell short in all critical measures by huge margins. *S. aegyptiacus* failed spectacularly by factors from four- to tenfold for maximum swimming speed on the surface or underwater (*Figure 3A*), for the capacity to right and remain stable or maneuver underwater (*Figure 3B*), and for generating the force needed to overcome buoyancy and fully submerge. *S. aegyptiacus* was an unstable, slow swimmer without the capacity to submerge. These are stiff biomechanical hurdles for the aquatic hypothesis to overcome.

We thought of other comparative means to test the aquatic hypothesis, plotting *S. aegyptiacus* against various extant and extinct secondarily aquatic amniotes to consider appendage area

(*Figure 5*), the size of foot and tail paddles in crocodylians (*Figure 6*), tail structure (*Figure 4C and D*), and the habitats occupied by large-bodied secondarily aquatic vertebrates (*Figure 7A*). *S. aegyptiacus* fails all of these comparative tests as well because it resembles other theropod dinosaurs in limb size, other reptiles that use midline sails for display, and semiaquatic reptiles in the diversity of coastal and inland habitats occupied.

Although additional fossils of *S. aegyptiacus* and other spinosaurids will surely come to light, the overall skeletal proportions and form of *S. aegyptiacus* are largely known. Although many fine points on the structure and function of this interesting clade of predators will surely continue to engender controversy. The aquatic hypothesis, for that reason, is unlikely to survive as a plausible lifestyle interpretation. What then is our lifestyle interpretation for *S. aegyptiacus*? Our study and that of *Hone and Holtz, 2021* envision *S. aegyptiacus* as a bipedal, semiaquatic dinosaur using ambush predation of large fish while wading into shallow coastal and riverine waters.

Thirteen principal conclusions can be drawn from this study, all of which may be tested:

1. Adult *S. aegyptiacus* had a body length of under 14 m with the axial column in neutral pose.
2. The reduced hind limb long bones in neotypic skeleton of *S. aegyptiacus* are infilled likely as an adaptation to weight support on land rather than functioning as ballast to increase density in water.
3. The segment-crossing caudal neural spines in *S. aegyptiacus* suggest that its tail functioned more as a pliant billboard than flexible fluke.
4. S. *aegyptiacus*, like S. *tenerensis* and other spinosaurids, was bipedal on land with its CM positioned over its hind feet. The long-clawed forelimbs of *S. aegyptiacus* were not used in weight support on land.
5. *S. aegyptiacus* could wade into shallow water for feeding with flotation occurring at water depth greater than ~2.6 m.
6. An adult flesh model of *S. aegyptiacus* has a body mass of ~7400 kg and average density of ~830 kg/m$^3$, which is considerably less than the density of saltwater (1026 kg/m$^3$).
7. *S. aegyptiacus* was incapable of diving, given its buoyancy and incompressible trunk. Full submergence would require 15–25 times the maximum force output of its tail, depending on estimated lung volume.
8. *S. aegyptiacus* was unstable in deeper water with little ability to right itself, swim, or maneuver underwater. Maximum power from its tail, assuming it could undulate as in *Alligator*, is less than 700 N, which would generate a top speed of ~1 m/s, an order of magnitude slower than extant large-bodied pursuit predators.
9. All extant and extinct large-bodied (>2 m long) secondarily aquatic vertebrates are strictly marine, whereas fossils pertaining to *Spinosaurus* have been found in inland basins distant from a marine coast.
10. Transition to a semiaquatic lifestyle, as occurred in the evolution of spinosaurid theropods, can occur at any body size. Transition to an aquatic lifestyle among tetrapods, in contrast, has only occurred at relatively small body size (<3 m) with subsequent radiation once in the marine realm into larger body sizes.
11. *S. aegyptiacus* is interpreted as a semiaquatic shoreline ambush predator more closely tied to waterways than baryonychine spinosaurids.
12. Spinosaurids flourished over a relatively brief Cretaceous interval (~35 My) in circum-Tethyan habitats with minimal impact on aquatic habitats globally.
13. Two phases are apparent in evolution of aquatic adaptations among spinosaurids, the second distinguishing spinosaurines as the most semiaquatic of non-avian dinosaurs.

## Materials and methods

### Skeletal reconstruction

The composite skeletal reconstruction of *S. aegyptiacus* is based principally on bones of holotypic and neotypic specimens supplemented by associated and isolated bones from Cenomanian-age formations in Egypt, Morocco, and Niger (*Figure 1A*). The two most important specimens include the subadult partial skeleton composing the holotype (BSP 9012 VIII 19) from the Western Desert of Egypt (*Stromer, 1915*; *Smith et al., 2006*) and a subadult partial skeleton designated as the neotype from

the Kem Kem Group in Morocco (FSAC-KK 11888; *Ibrahim et al., 2014*; *Ibrahim et al., 2020b*). A third referred specimen from Egypt was also considered (BSPG 1922 X45, '*Spinosaurus* B'; *Stromer, 1934*). These are the only associated specimens known for *S. aegyptiacus* on which to base the skeletal reconstruction, the relative size calculated from overlapping bones (*Table 1*). Of these three specimens, only the bones of the neotype are preserved, all of which have been CT-scanned except for recently discovered bones of the tail (*Ibrahim et al., 2020b*). Noteworthy isolated specimens have been recovered from the Kem Kem Group in Morocco, including a large snout and manual phalanx used to gauge maximum adult body size.

We incorporated all CT-scanned bones of the neotype and reconstructions (based on lithographic plates and photographs) of bones of the holotype and referred specimen from Egypt. For unknown bones without sequential adjacency as a guide, other spinosaurids were consulted for shape and proportion. All digital bones were articulated in osteologically neutral pose (*Stevens and Parrish, 1999*), a standardizing criterion for comparing vertebrate skeletons (*Mallison, 2010*; *Vidal et al., 2020*). In the case of overlapping bones, priority was given to the neotype (blue) followed by the holotype (red) and referred specimens (yellow). Bones without representation among specimens attributed to *S. aegyptiacus* are shaded gray (*Figure 1A*).

## Skeletal reconstructions compared

We compared our digital skeletal model of *S. aegyptiacus* to the recently published 2D silhouette skeletal reconstruction in the aquatic hypothesis (*Ibrahim et al., 2020b*), both of which are based primarily on holotypic and neotypic specimens (*Figure 9*). We registered the reconstructions to each other by superimposing the four longest complete bones of the neotype (femur, tibia, ilium, ischium). Significant differences are apparent in several dimensions with major implications for the calculation of CM and CB.

When aligned at the hip, sacral and caudal columns have nearly identical length, but the presacral column is significantly longer (~10%) in the reconstruction of the aquatic hypothesis. The extra length of the presacral column is located in the neck between C2-10 and torso between D4-13. The trunk in our digital skeletal model is also not as deep as that in skeletal silhouette drawing, as can be seen by aligning the skeletons along the dorsal column (*Figure 9C*). The contour of the belly marked by the gastral basket and the coracoids of the pectoral girdle extend farther ventrally (~25%) than the ends of the pubes, unlike our digital reconstruction or that of most other silhouette reconstructions for

---

**Table 1.** Relative size of specimens in the skeletal reconstruction of *S. aegyptiacus*.
Relative sizes of key specimens used in the skeletal model of *S. aegyptiacus* (nos. 1–4) and select bones (nos. 5, 6) from Egypt and Morocco. All are scaled to the size of the adult snout (MSMN V4047).

| No. | Specimen | Maturity | Relative size (%) | Linear upsizing | Description |
|---|---|---|---|---|---|
| 1 | BSPG 1912 VIII 19 | Subadult | 76 | 1.32 | Holotype (destroyed) preserving dentaries, presacral, sacral and caudal vertebrae including the dorsal sail (*Stromer, 1915*) |
| 2 | FSAC-KK 11888 | Subadult | 76 | 1.32 | Neotype preserving skull bones, partial limbs, dorsal sail and most of the tail (*Ibrahim et al., 2014*) |
| 3 | BSPG 1922X45 | Subadult | 66 | 1.51 | '*Spinosaurus* B' (destroyed) fragmentary specimen with five partial dorsals (~D1 centrum, mid dorsal centrum, partial ~D13 vertebra), seven partial caudal vertebrae, and both tibiae (*Stromer, 1934*) |
| 4 | MSMN V4047 | Adult | 100 | — | Isolated snout with broken teeth (*Dal Sasso et al., 2005*); large size and coossified sutures indicate maturity |
| 5 | UCRC PV8 | Adult | 80 | 1.25 | Large manual phalanx I-1 (28.0 cm length) of an adult within reach but still smaller than the 35 cm length estimated on the basis of the proportions in the manus of *Angaturama* (?=*Irritator*) scaled to the adult snout (*Aureliano et al., 2018*) |
| 6 | UCRC PV24 | Adult | 75 | 1.33 | Large Kem Kem vertebra, ~C9, from Gara Sbaa (centrum length 11.6 cm, centrum width 14.0 cm) |

non-avian theropods. The length of the ribcage in our model is consistent with the only well-preserved spinosaurid ribcage known to date (*S. tenerensis,* MNBH GAD70).

Finally, the forelimb in the skeletal silhouette drawing is ~30% longer than that in our digital reconstruction (*Figure 9A*). The neotype is the only associated specimen of *S. aegyptiacus* preserving bones from the forelimb (partial manual digit II). The preserved manual phalanges are slender with deeply cleft distal condyles, which allows reference of additional phalanges of similar form from the Kem Kem Group (*Figure 1D*). Our reconstruction of the manus is based on a recently described forelimb of the close relative *Irritator* (=*Angaturama*; *Machado and Kellner, 2009*; *Aureliano et al., 2018*). The proportions of more proximal forelimb segments and the pectoral girdle are based on the holotypic specimens of *Baryonyx* and *Suchomimus*. The forelimb in *S. aegyptiacus* is robust and long relative to other non-avian theropods, although considerably shorter than in some previous reconstructions (*Ibrahim et al., 2014*).

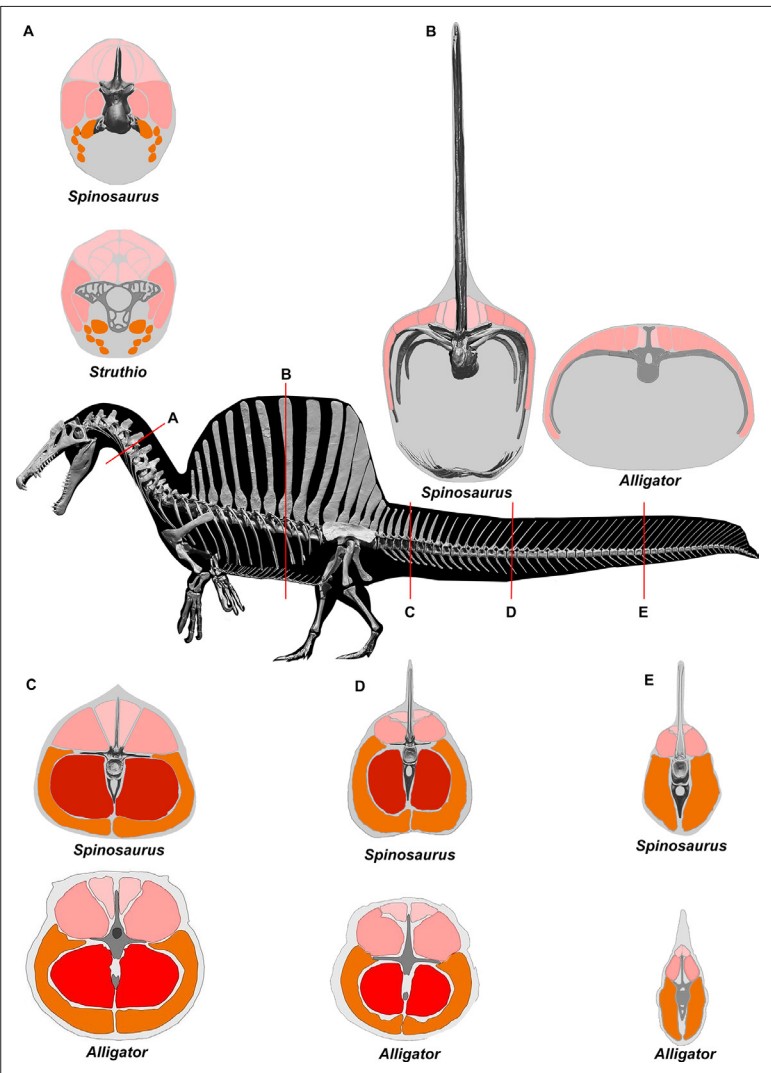

**Figure 10.** CT scans inform cross-sectional muscle mass in *Spinosaurus aegyptiacus*. Muscle mass reconstructions of the axial column at five points (**A-E**) in *S. aegyptiacus* are compared to CT scan cross-sections of *Struthio* (*Snively and Russell, 2007*) and *Alligator* (*Wedel, 2003*; *Mallison et al., 2015*).

The longer presacral proportions, deeper torso, and longer forelimb of the skeletal reconstruction and flesh model used by the aquatic hypothesis cantilever show significant additional body mass anterior to the hip joint. That additional front-loading appears to be the main factor generating their mid trunk location for CM (*Ibrahim et al., 2020b*), which constitutes the basis for regarding *S. aegyptiacus* as a quadruped on land (*Ibrahim et al., 2014*).

## Flesh reconstruction of axial musculature

To estimate the volume of axial musculature in *S. aegyptiacus* (*Figure 10*), we referenced CT-based studies on the ostrich (*Struthio*; *Wedel, 2003*; *Snively and Russell, 2007*; *Persons et al., 2020*) and alligator (*Alligator*; *Cong et al., 1998*; *Mallison et al., 2015*). To estimate caudal muscle mass, we used CT scans of various reptiles including the sail-backed basilisk lizard, *Basiliscus plumifrons* (*Figure 11*, *Tables 2–5*).

For *epaxial muscle mass* in *S. aegyptiacus*, we estimated its vertical extent as twice centrum height, measuring upward from the base of the neural spine. The transverse width of epaxial musculature was estimated to be a little less than that of the hypaxial muscles, widest ventrally and tapering to the midline dorsally. For *hypaxial muscle mass* in *S. aegyptiacus*, we estimated its vertical depth at approximately twice chevron length in the anterior tail and 1.5 times chevron length in mid and posterior portions of the tail. We estimated the transverse width of hypaxial muscles as twice the length of the transverse processes. CT cross-sections of extant reptiles show that considerable muscle mass is present beyond the distal end of caudal transverse processes in anterior and middle portions of the tail (*Figure 10*).

Several cross-sections from *Basiliscus plumifrons* (crested basilisk) provided valuable insights on the distribution of axial muscles in a lizard with a dorsal-to-caudal sail (*Figure 11*, *Tables 2–5*). Epaxial musculature in the trunk and tail comprises less than one-third of total axial muscle volume (*Table 2*). Caudal neural spines project beyond the epaxial musculature to support the sail to a greater extent in mid and distal portions of the tail. At the base of the tail (CA4), approximately one-third (29.3%) of the neural spine projects dorsally supporting the sail. At mid tail (CA15), approximately three-quarters

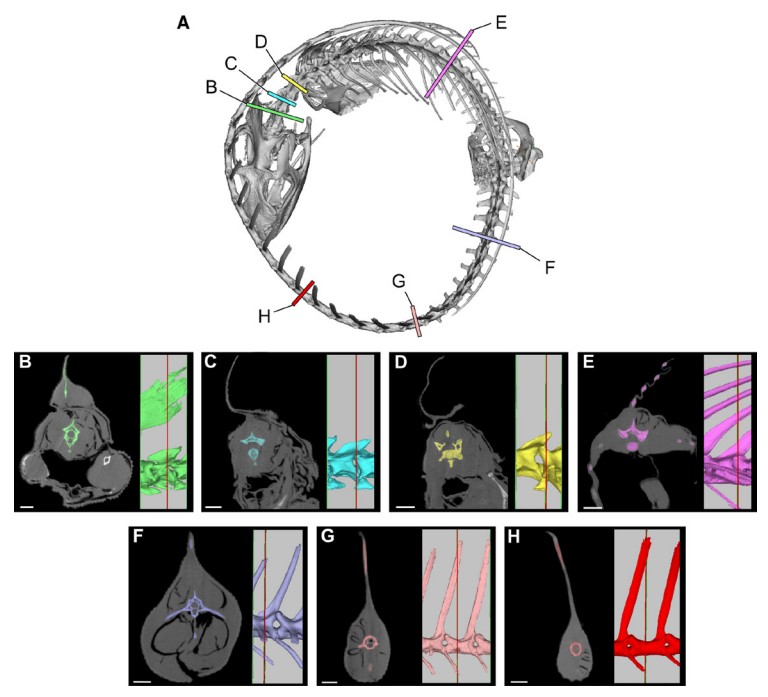

**Figure 11.** Cross-sections from a CT scan of *Basiliscus plumifrons* FMNH 112993. (**A**) Skeleton showing position of CT sections of the axial column. (**B**) Anterior cervical region (C2). (**C**) Mid cervical region (C3). (**D**) Posterior cervical and anterior dorsal region (C4-D1). (**E**) Mid dorsal region (D12). (**F**) Anterior caudal region (CA4). (**G**) Caudal region at most posterior transverse process (CA10). (**H**) Mid caudal region (CA15). CT scan data available on Morphosource.org. Scale bars, 5 mm.

**Table 2.** Axial muscle area in the crested basilisk.

Area measurements of epaxial and hypaxial musculature along the axial column in the crested basilisk *Basiliscus plumifrons* (FMNH 112993). C, cervical; D, dorsal; CA, caudal.

| Location | Total area | Epaxial area | | Hypaxial area | |
|---|---|---|---|---|---|
| | mm² | mm² | % total | mm² | % total |
| Anterior neck (C2) | 180.4 | 83.9 | 46.5 | 96.5 | 53.5 |
| Mid neck (C3) | 139.7 | 54.9 | 39.3 | 84.8 | 60.7 |
| Posterior neck (C4-D1) | 202.1 | 69.8 | 34.7 | 132.3 | 65.3 |
| Mid dorsal (D12) | 212.0 | 60.0 | 28.3 | 152.0 | 71.7 |
| Basal tail (CA4) | 434.0 | 153.8 | 35.4 | 280.2 | 64.6 |
| Anterior tail (CA10, last transverse process) | 221.7 | 63.5 | 28.6 | 158.2 | 71.4 |
| Mid tail (CA15) | 97.8 | 29.8 | 29.8 | 68.0 | 70.2 |

(76.8%) of the neural spine projects dorsally supporting the sail. The hypaxial musculature extends well below the distal end of the chevrons. At the base of the tail (CA4), the chevron lies internal to approximately one-half (52.6%) of hypaxial muscle depth, with the remainder (47.4%) distal to the end of the chevon. Farther along the tail (CA10-15), the chevrons are proportionately longer, supporting approximately two-thirds (67%) of hypaxial muscle depth with approximately one-third of hypaxial musculature beyond the distal end of the chevron. These cross-sections confirm the presence of considerable muscle mass ventral to the distal end of the chevrons in anterior through mid portions of the tail (*Figures 10 and 11*).

## Flesh model density partitions, dimensions, and properties

The digital skeletal model was wrapped in flesh (using ZBrush) guided by recent documentation of muscle mass in CT scans of various modern analogs (*Wedel, 2003*; *Snively and Russell, 2007*; *Mallison et al., 2015*; *Persons et al., 2020*; *Díez Díaz et al., 2020*). We inserted anatomically shaped and -positioned air spaces (pharynx-trachea, paraxial air sacs, lungs) of optional volumes (minimum-lizard, medium-crocodylian, maximum-avian) within the head, neck, and torso (*Figure 2C*). For additional measurements, we added a 'mesh' over the flesh model (*Henderson, 1999*; *Figure 2D*).

We divided the flesh model into six parts (axial body-presacral, axial-caudal, dorsal-to-caudal sail, forelimbs, hindlimbs, lungs) in order to assign appropriate densities (*Table 6*). Densities were assigned to body parts based on their estimated composition, using values for tissues ranging from fat (900 g/l) to compact bone (2000 g/l). The average whole-body density for *S. aegyptiacus*, 833 g/l (*Table 6*) compares favorably to whole-body density estimates for various non-avian dinosaurs (800–900 g/l). We have compiled various functional dimensions of the adult flesh model (*Table 7*), and we divided

**Table 3.** Axial muscle and transverse process length in the tail of the crested basilisk.

Transverse processes versus muscle width in the tail cross-sections in the crested basilisk *Basiliscus plumifrons* (FMNH 112993). Measurements are from the midline to the distal end of the transverse process (or centrum margin when there is no process) and to the lateral surface of the tail. CA, caudal.

| Location | Measurement | Total (mm) | % bone and muscle | % muscle only |
|---|---|---|---|---|
| | Transverse process width | 16.8 | | |
| CA4 | Total width | 26.7 | 62.9 | 37.1 |
| | Transverse process width | 8.1 | | |
| CA10 | Total width | 14.2 | 57 | 43.0 |
| | Centrum width | 3.5 | | |
| CA15 | Total width | 9.5 | 36.8 | 63.2 |

**Table 4.** Epaxial muscle height and neural spine height in the tail of the crested basilisk.
Height of neural spines versus epaxial musculature in tail cross-sections in the crested basilisk
*Basiliscus plumifrons* (FMNH 112993). Measurements are from the dorsal surface of the centrum to
the top of the epaxial muscle and to the distal end of the neural spine. CA, caudal.

| Location | Measurement | Total (mm) | % neural spine adjacent to muscle | % neural spine above muscle |
|---|---|---|---|---|
| | Neural spine height | 18.4 | | |
| CA4 | Epaxial mm height | 13.0 | 70.7 | 29.3 |
| | Neural spine height | 33.6 | | |
| CA10 | Epaxial mm height | 10.8 | 32.1 | 67.9 |
| | Neural spine height | 34.5 | | |
| CA15 | Epaxial mm height | 8.0 | 23.2 | 76.8 |

the flesh model into 10 body parts, for which we list volumes and external surface areas (excluding cut surfaces) (*Table 8*).

We registered CM as the horizontal distance from the apex of the acetabulum (x-coordinate) and the vertical distance from the ground surface under the sole of the foot (y-coordinate) (*Table 9*, no. 4). This is the fourth estimation of CM for *S. aegyptiacus*, and we argue here the most accurate. We prefer a measure from the acetabulum rather than the distal tail tip, which as in *S. aegyptiacus* is often a matter of speculation given the rarity of completely preserved caudal columns (*Hone, 2012*). For the acetabulum, we recommend using its 'apex' rather than its 'cranial end' (*Ibrahim et al., 2020b*) for three reasons. First, the apex of the acetabulum is a more easily recognized landmark than the poorly defined anterior edge (or rim) of the acetabulum. Second, the apex rather than the 'cranial end' of the acetabulum is a more functionally intuitive point from which to measure CM, given its proximity to the rotational point for body mass centered over the hind limbs. And third, the dorsal (proximal) articular end of the femoral head is close to the apex of the acetabulum, and so the length of the femur and the distance that CM lies farther forward can be directly compared (CM located anteriorly beyond femoral length excludes stable bipedal posture with a relatively horizontal dorsosacral column).

With avian-like air space ('maximum'), CM is positioned only 15.3 cm anterior to the apex of the acetabulum and clearly over the pedal phalanges of the foot for a bipedal stance (*Table 10*). The smallest air space option modeled on lizards ('minimum,' only 4% of body volume) generates the heaviest torso and displaces CM anteriorly 13.2 cm to a distance of 28.5 cm from the apex of the acetabulum (*Table 10*). In this location, CM is still ~12 cm short of the midpoint along the length of the femur (~40 cm; femoral length is 81 cm in adult *S. aegyptiacus*). In this worst-case scenario regarding internal air volume, CM is still positioned over the pedal phalanges of the hind limb. Our flesh model does not support an obligatory quadrupedal pose on land for *S. aegyptiacus*.

**Table 5.** Hypaxial muscle depth and chevron length in the tail of the crested basilisk.
Chevron length versus hypaxial muscle depth in tail cross-sections in the crested basilisk *Basiliscus plumifrons* (FMNH 112993). Measurements are from the ventral surface of the centrum to the distal tip of the chevron and to the ventral surface of the tail. CA, caudal.

| Location | Measurement | Total (mm) | % chevron length | % muscle below chevron |
|---|---|---|---|---|
| | Chevron depth | 9.0 | | |
| CA4 | Hypaxial depth | 17.1 | 52.6 | 47.4 |
| | Chevron depth | 7.1 | | |
| CA10 | Hypaxial depth | 10.6 | 67.0 | 33.0 |
| | Chevron depth | 4.8 | | |
| CA15 | Hypaxial depth | 7.1 | 67.6 | 32.4 |

**Table 6.** Density, volume, and mass in the flesh model of *S. aegyptiacus*.
Whole-body and body part densities, volumes, and masses for the new mesh adult flesh model of *S. aegyptiacus*.

| No. | Body partition | Average density (kg/m³) | % of axial volume without sail | Mass (kg) |
|---|---|---|---|---|
| 1 | Whole body | 833 | — | 7390 |
| 2 | Axial body (excluding lung/sail) | 788 | 100.0 | 5794 |
| 3 | Axial head-trunk (not lung/sail/tail) | 850 | 64.8 | 3209 |
| 4 | Axial tail (not sail) | 1000 | 35.2 | 2585 |
| 5 | Forelimb (paired) | 1050 | 3.8 | 108 |
| 6 | Hind limb (paired) | 1050 | 11.1 | 590 |
| 7 | Dorsocaudal sail | 1196 | 8.5 | 441 |
| 8 | Lungs | 0 | 12.5 | 0 |

**Table 7.** Flesh model functional dimensions in *S. aegyptiacus*.
Functional dimensions for the adult flesh model of *S. aegyptiacus* in sculling pose.

| No. | Dimension | Measure (m) |
|---|---|---|
| 1 | Total body length (sculling pose) | 13.53 |
| 2 | Body length minus tail (sculling pose) | 6.92 |
| 3 | Head length | 1.57 |
| 4 | Neck length (sculling pose) | 2.18 |
| 5 | Trunk depth (mid trunk, without sail) | 1.28 |
| 6 | Trunk sail depth (mid trunk) | 1.93 |
| 7 | Trunk sail length (maximum at base) | 3.53 |
| 8 | Tail length | 6.61 |
| 9 | Tail depth at base | 1.38 |
| 10 | Tail depth at midpoint | 0.97 |
| 11 | Tail depth at distal end | 0.87 |
| 12 | Tail depth average | 1.08 |
| 13 | Forelimb length (straightened) | 1.85 |
| 14 | Hind limb length (straightened) | 2.88 |

**Table 8.** Flesh model volume and surface area in *S. aegyptiacus*.

Adult flesh model whole-body and body part volumes and surface areas as measured in MeshLab. Surface area of body parts does not include cut surfaces.

| No. | Body part | Volume (m³) | Surface area (m²) |
|---|---|---|---|
| 1 | Whole body | 8.94 | 54.06 |
| 2 | Body above waterline (floating) | 1.65 | 22.58 |
| 3 | Body below waterline (floating) | 7.27 | 31.38 |
| 4 | Head | 0.21 | 2.23 |
| 5 | Neck | 0.78 | 4.37 |
| 6 | Trunk | 4.01 | 11.44 |
| 7 | Trunk sail (both sides, external edge) | 0.40 | 10.06 |
| 8 | Forelimb (both) | 0.24 | 3.86 |
| 9 | Hind limb (both) | 0.45 | 5.29 |
| 10 | Tail | 2.81 | 16.56 |
| 11 | Tail with axial muscle | 2.71 | 13.27 |
| 12 | Tail sail only | 0.10 | 3.17 |
| 13 | Airspace-minimum (~4% body volume) | 0.37 | 4.86 |
| 14 | Airspace-medium (~8% body volume) | 0.67 | 6.63 |
| 15 | Airspace-maximum (~12% body volume) | 1.08 | 8.66 |

**Table 9.** Center of mass (CM) calculations for *S. aegyptiacus*.

There have been four estimates for the location of CM in flesh models of *S. aegyptiacus* using four different points of origin as a reference. Because three were based on an adult flesh model, we convert the one study based on a subadult (number 3) to reflect its position in an adult flesh model: "x" is the distance anterior to the origin, and "y" is the height above the ground, both in cm.

| No. | Author/result | x-origin | x | y | Notes |
|---|---|---|---|---|---|
| 1 | ***Ibrahim et al., 2014***; *quadruped* | Hip joint | >81 | – | Based on an adult flesh model, no coordinates given, CM shown graphically under D10 and said to be anterior to hip/knee joints at a distance greater than femur length (MeshLab calculation error) |
| 2 | ***Henderson, 2018***; *biped* | Tip of tail | 8,850 | 100 | Based on an 3D mesh model based on the adult skeletal model of ***Ibrahim et al., 2014*** with estimated length 16 m long; y-coordinate origin is 'lowest point of axial body' |
| 3 | ***Ibrahim et al., 2020b***; *quadruped* | 'Cranial rim' of acetabulum | 72.5–82.5 (adult = 95.7–108.9) | –81 (adult: 106.9) | Based on a flesh model of the subadult neotype with femur length of 62.5 cm (actual 61.0 cm); y-coordinate measures to substrate |
| 4 | This paper; *biped* | Apex of the acetabulum | 15.3 | –240 | Based on an adult flesh model with avian-style internal air spaces and femur length of 81.0 cm; y-coordinate measures to substrate |

**Table 10.** Estimated internal air space in *S. aegyptiacus*.
Air space options for the adult flesh model of *S. aegyptiacus* and their effect on whole-body density, body mass (BM), and center of mass (CM). The x-coordinate for CM is measured from the apex of the acetabulum.

| No. | Air space option | Part of whole-body volume (%) | Mean whole-body density (g/l) | BM (kg) | CM x-coordinate (cm) |
|---|---|---|---|---|---|
| 1 | Minimum (lizard-like) | 4.0 | 909 | 8013 | 28.5 |
| 2 | Medium (croc-like) | 8.0 | 875 | 7716 | 23.2 |
| 3 | Maximum (bird-like) | 12.5 | 833 | 7390 | 15.3 |

# Acknowledgements

We thank A Resetar, J Mata, and D Coldren of The Field Museum, R Carmichael of the Wildlife Center and R Bavrisha of the Chicago Herpetological Society for access to recent preserved and live reptile specimens, MorphoSource, FMNH, UMMZ, UF, and KU for access to and archiving of CT scans, R Shonk and A Schulte for digital modeling, E Fitzgerald for fossil preparation, J Mallon for photography, and J Schwartz, E Saitta, E Johnson-Ransom, DB Dutheil, SW Evers, E Snively, D Hone, and T Holtz for comments on the manuscript. This research was supported by Bob and Ellen Vladem and SC Johnson.

# Additional information

## Funding

| Funder | Grant reference number | Author |
|---|---|---|
| Bob and Ellen Vladem | Gift | Paul C Sereno |
| SC Johnson | Gift | Paul C Sereno |
| Chicago Herpetological Society | | Paul C Sereno |

The funders had no role in study design, data collection and interpretation, or the decision to submit the work for publication.

## Author contributions

Paul C Sereno, Conceptualization, Resources, Formal analysis, Supervision, Funding acquisition, Validation, Investigation, Visualization, Writing - original draft, Project administration, Writing – review and editing; Nathan Myhrvold, Formal analysis, Validation, Investigation, Visualization, Methodology, Writing – review and editing; Donald M Henderson, Frank E Fish, Conceptualization, Formal analysis, Investigation, Visualization, Methodology, Writing – review and editing; Daniel Vidal, Conceptualization, Formal analysis, Investigation, Visualization, Methodology, Writing – review and editing, phylogenetic analysis; Stephanie L Baumgart, Conceptualization, Formal analysis, Investigation, Visualization, Methodology, Writing – review and editing, CT data on recent organisms; Tyler M Keillor, Data curation, Formal analysis, Visualization, Methodology, Writing – review and editing, CT-based model reconstruction; Kiersten K Formoso, Conceptualization, Investigation, Visualization, Methodology, Writing – review and editing; Lauren L Conroy, Data curation, Investigation, Visualization, Visualisations

## Author ORCIDs

Paul C Sereno  http://orcid.org/0000-0001-7958-3701
Daniel Vidal  http://orcid.org/0000-0002-6054-1357
Stephanie L Baumgart  http://orcid.org/0000-0001-9534-7389

**Decision letter and Author response**
Decision letter https://doi.org/10.7554/eLife.80092.sa1
Author response https://doi.org/10.7554/eLife.80092.sa2

## Additional files

### Supplementary files
- MDAR checklist
- Source data 1. Data matrix as nex file for Mesquite.
- Source data 2. Data matrix as tnt file for TNT.

### Data availability

All data generated or analysed during this study are included in the manuscript and appendices. Data has been deposited to MorphoSource and provided as Source datas 1 and 2.

The following previously published dataset was used:

| Author(s) | Year | Dataset title | Dataset URL | Database and Identifier |
|---|---|---|---|---|
| Sereno PC, Myrhvold N, Henderson DM, Fish FE, Vidal D, Baumgart SL, Keillor TM, Formoso KK, Conroy LL | 2020 | Spinosaurus was not an aquatic dinosaur | https://www.morphosource.org/projects/000460619 | MorphoSource, 000460619 |

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

## Appendix 1

### Long bone infilling in *S. aegyptiacus*

In *S. aegyptiacus,* the medullary cavities in the long bones of the hind limb are reduced in diameter or infilled altogether by non-pachystotic bone. This condition is unusual among non-avian dinosaurs and was initially thought to be an adaptation for decreasing buoyancy (*Ibrahim et al., 2014*; *Fabbri et al., 2022*). Enhanced bone strength is an alternative explanation for solid hind limb long bones in *S. aegyptiacus*, a very large bipedal theropod with reduced hind limbs. Similar long bone infilling occurs in other large-bodied bipedal and quadrupedal dinosaurs and mammals (*Vanderven et al., 2014*; *Houssaye et al., 2016*).

To better understand this alternative explanation, we compare the increase in strength of a femur with a solid shaft compared to one of identical size with a hollow shaft using the section modulus (SM) as an indicator of the resistance to bending (*Farlow et al., 1995*). For a solid circular cross-section, the SM is given by $\frac{\pi}{4}R^3$ , where R is the outer radius. When R has a value 1.0, SM is $\frac{\pi}{4}$ , or 0.785 m³. The SM for a cylinder with a hollow core is calculated by subtracting the SM of an inner cylinder comprising the hollowed space from that of the outer cylinder.

An inner radius of 0.5 generates a hollow cylinder that decreases the solid cylinder volume with 25% and generates an SM of $\frac{\pi}{4}\left(1.0^3 - 0.5^3\right) = \frac{\pi}{4}\left(0.875\right)$ . This is a reduction of ~13% of SM. However, many theropods have hollow cores within long bones that comprise 50% or more of the volume of the long bone. For a hollow core equal to 50% of outer cylinder volume, the inner radius is 0.707. The SM, calculated by subtracting SM for a cylinder of radius 0.707 from the full cylinder of radius 1.0, is $\frac{\pi}{4}\left(1.0^3 - 0.707^3\right)$ , or $\frac{\pi}{4}\left(0.646\right)$ , a reduction of ~35% of the SM of a solid cylinder.

Thus, infilling of hind limb bones in *S. aegyptiacus* may have increased bending strength by as much as 35%. The femur in *S. aegyptiacus*, in particular, may have been subjected to substantial bending forces. The hypertrophied attachment flange for the caudofemoralis muscle occupies almost one-third of the length of the femoral shaft.

# Appendix 2

## Extant and extinct comparative materials

We used references in the literature, extant amphibian and reptile specimens, and a mosasaur to estimate muscle mass in the flesh reconstruction of *S. aegyptiacus* and to measure caudal centrum proportions along the tail (*Figures 2 and 4D*, *Appendix 2—table 1* and *Appendix 2—table 2*).

**Appendix 2—table 1.** Extant newt and squamate specimens used for muscle mass estimation and for logging centrum proportions along the tail.

Specimen numbers have hyperlinks to MorphoSource specimen pages with associated scans and 3D models.

| Specimen no. | Taxon | Sex | Type | Purpose |
|---|---|---|---|---|
| FMNH 84926 | *Triturus cristatus* | Male | Wet | Caudal centrum ratios, muscle mass estimation |
| FMNH 57512 | *Intellagama lesueurii* | Male | Wet | Muscle mass estimation |
| FMNH 22389 | *Intellagama lesueurii* | Male | Skeleton | Caudal centrum ratios, muscle mass estimation |
| KU 314941 | *Hydrosaurus amboinensis* | Unknown; low neural spines (female/ immature male) | Wet | Caudal centrum ratios |
| FMNH 52698 | *Hydrosaurus pustulatus* | Male | Wet | Muscle mass estimation |
| FMNH 236131 | *Hydrosaurus pustulatus* | Unknown; low neural spines (female/ immature male) | Skeleton | Muscle mass estimation |
| FMNH 112993 | *Basiliscus plumifrons* | Male | Wet | Muscle mass estimation |
| FMNH 257162 | *Basiliscus plumifrons* | Male | Skeleton | Additional high-resolution visualization |
| UMMZ 121461 | *Basiliscus basiliscus* | Male | Wet | Caudal centrum ratios |
| UF 41558 | *Amblyrhynchus cristatus* | Unknown | Wet | Caudal centrum ratios |
| FMNH 22042 | *Amblyrhynchus cristatus* | Unknown;?male (tallest caudal neural spines in FMNH collection) | Skeleton | Muscle mass estimation |
| UF 21461 | *Alligator mississippiensis* | Unknown; juvenile | Wet | X-ray image of caudal centra in tail; caudal centrum ratios |

**Appendix 2—table 2.** Caudal centrum ratios along the tail in *S. aegyptiacus*, *Mosasaurus*, and extant semiaquatic amphibians and reptiles (*Figure 4D*).

| CA # | L-H ratios | | | | | | | | Relative distance (caudal #/total # of caudals) | | | | | | | |
|---|---|---|---|---|---|---|---|---|---|---|---|---|---|---|---|---|
| | Spino. | Mosa. | Ambly. | Alli. | Basil. | Hydro. | Trit. | Intel. | Spino. | Mosa. | Ambly. | Alli. | Basil. | Hydro. | Trit. | Intel. |
| 1 | 0.73 | | 1.40 | 1.14 | 1.52 | 1.25 | 3.26 | 1.60 | 0.02 | 0.01 | 0.02 | 0.02 | 0.02 | 0.02 | 0.03 | 0.02 |
| 2 | | | 1.56 | 1.11 | 1.92 | 1.50 | 3.78 | 2.21 | 0.04 | 0.03 | 0.04 | 0.05 | 0.04 | 0.04 | 0.05 | 0.04 |
| 3 | | | 1.55 | 1.05 | 2.05 | 1.97 | 3.18 | 2.48 | 0.06 | 0.04 | 0.06 | 0.07 | 0.06 | 0.05 | 0.08 | 0.06 |
| 4 | 0.84 | | 1.88 | 1.18 | 2.30 | 2.05 | 3.20 | 2.73 | 0.08 | 0.05 | 0.09 | 0.10 | 0.08 | 0.07 | 0.11 | 0.08 |
| 5 | | | 1.86 | 1.22 | 2.44 | 2.14 | 2.88 | 2.78 | 0.10 | 0.07 | 0.11 | 0.12 | 0.10 | 0.09 | 0.14 | 0.10 |
| 6 | | | 1.96 | 1.33 | 2.50 | 2.14 | 3.08 | 2.83 | 0.12 | 0.08 | 0.13 | 0.15 | 0.12 | 0.11 | 0.16 | 0.13 |
| 7 | 1.12 | 1.06 | 2.06 | 1.38 | 2.55 | 2.26 | 3.05 | 2.98 | 0.14 | 0.09 | 0.15 | 0.17 | 0.14 | 0.13 | 0.19 | 0.15 |
| 8 | 1.15 | 1.03 | 2.10 | 1.43 | 2.72 | 2.17 | 3.11 | 2.99 | 0.16 | 0.11 | 0.17 | 0.20 | 0.16 | 0.14 | 0.22 | 0.17 |
| 9 | | 0.95 | 2.13 | 1.49 | 2.87 | 2.41 | 3.07 | 3.22 | 0.18 | 0.12 | 0.19 | 0.22 | 0.18 | 0.16 | 0.24 | 0.19 |
| 10 | | 0.97 | 2.19 | 1.52 | 2.89 | 2.45 | 2.90 | 3.26 | 0.20 | 0.14 | 0.21 | 0.24 | 0.20 | 0.18 | 0.27 | 0.21 |
| 11 | | 0.98 | 2.29 | 1.53 | 3.14 | 2.34 | 2.95 | 3.17 | 0.22 | 0.15 | 0.23 | 0.27 | 0.22 | 0.20 | 0.30 | 0.23 |

*Appendix 2—table 2 Continued on next page*

*Appendix 2—table 2 Continued*

| | L-H ratios | | | | | | | | Relative distance (caudal #/total # of caudals) | | | | | | | |
|---|---|---|---|---|---|---|---|---|---|---|---|---|---|---|---|---|
| CA # | Spino. | Mosa. | Ambly. | Alli. | Basil. | Hydro. | Trit. | Intel. | Spino. | Mosa. | Ambly. | Alli. | Basil. | Hydro. | Trit. | Intel. |
| 12 | | 0.93 | 2.37 | 1.67 | 2.94 | 2.50 | 3.17 | 3.28 | 0.24 | 0.16 | 0.26 | 0.29 | 0.24 | 0.21 | 0.32 | 0.25 |
| 13 | 1.24 | 1.07 | 2.57 | 1.62 | 3.27 | 2.48 | 2.73 | 3.34 | 0.26 | 0.18 | 0.28 | 0.32 | 0.25 | 0.23 | 0.35 | 0.27 |
| 14 | 1.21 | 1.04 | 2.52 | 1.65 | 3.35 | 2.43 | 2.77 | 3.21 | 0.28 | 0.19 | 0.30 | 0.34 | 0.27 | 0.25 | 0.38 | 0.29 |
| 15 | | 1.06 | 2.59 | 1.74 | 3.49 | 2.64 | 2.85 | 3.51 | 0.30 | 0.20 | 0.32 | 0.37 | 0.29 | 0.27 | 0.41 | 0.31 |
| 16 | | 0.91 | 2.67 | 1.72 | 3.46 | 2.52 | 2.95 | 3.64 | 0.32 | 0.22 | 0.34 | 0.39 | 0.31 | 0.29 | 0.43 | 0.33 |
| 17 | 1.43 | 0.92 | 2.74 | 1.91 | 3.55 | 2.94 | 2.65 | 3.52 | 0.34 | 0.23 | 0.36 | 0.41 | 0.33 | 0.30 | 0.46 | 0.35 |
| 18 | 1.33 | 0.95 | 2.80 | 1.81 | 3.66 | 2.82 | 3.29 | 3.73 | 0.36 | 0.24 | 0.38 | 0.44 | 0.35 | 0.32 | 0.49 | 0.38 |
| 19 | | 0.99 | 2.91 | 1.89 | 3.73 | 3.03 | 3.14 | 3.61 | 0.38 | 0.26 | 0.40 | 0.46 | 0.37 | 0.34 | 0.51 | 0.40 |
| 20 | 1.37 | 0.98 | 2.77 | 1.97 | 3.89 | 2.85 | 2.70 | 3.50 | 0.40 | 0.27 | 0.43 | 0.49 | 0.39 | 0.36 | 0.54 | 0.42 |
| 21 | 0.97 | 0.99 | 2.94 | 1.89 | 3.91 | 2.89 | 2.82 | 3.73 | 0.42 | 0.28 | 0.45 | 0.51 | 0.41 | 0.38 | 0.57 | 0.44 |
| 22 | 1.40 | 0.98 | 2.99 | 1.95 | 3.99 | 2.87 | 3.16 | 3.69 | 0.44 | 0.30 | 0.47 | 0.54 | 0.43 | 0.39 | 0.59 | 0.46 |
| 23 | 1.38 | 1.02 | 2.99 | 2.12 | 4.01 | 2.82 | 3.27 | 3.93 | 0.46 | 0.31 | 0.49 | 0.56 | 0.45 | 0.41 | 0.62 | 0.48 |
| 24 | 1.32 | 0.92 | 3.00 | 2.21 | 3.74 | 2.96 | 2.92 | 3.90 | 0.48 | 0.32 | 0.51 | 0.59 | 0.47 | 0.43 | 0.65 | 0.50 |
| 25 | 1.41 | 0.88 | 3.05 | 2.25 | 4.40 | 3.29 | 3.22 | 3.96 | 0.50 | 0.34 | 0.53 | 0.61 | 0.49 | 0.45 | 0.68 | 0.52 |
| 26 | 1.36 | 0.90 | 3.05 | 2.49 | 4.14 | 3.12 | 3.63 | 3.72 | 0.52 | 0.35 | 0.55 | 0.63 | 0.51 | 0.46 | 0.70 | 0.54 |
| 27 | 1.35 | 0.93 | 3.02 | 2.63 | 4.25 | 3.31 | 3.45 | 3.99 | 0.54 | 0.36 | 0.57 | 0.66 | 0.53 | 0.48 | 0.73 | 0.56 |
| 28 | 1.15 | 0.90 | 3.24 | 2.79 | 3.69 | 3.13 | 3.36 | 4.03 | 0.56 | 0.38 | 0.60 | 0.68 | 0.55 | 0.50 | 0.76 | 0.58 |
| 29 | 1.44 | 0.83 | 3.27 | 2.70 | 4.37 | 3.25 | 3.03 | 4.14 | 0.58 | 0.39 | 0.62 | 0.71 | 0.57 | 0.52 | 0.78 | 0.60 |
| 30 | 1.29 | 0.89 | 3.31 | 2.99 | 3.97 | 3.18 | 3.11 | 4.06 | 0.60 | 0.41 | 0.64 | 0.73 | 0.59 | 0.54 | 0.81 | 0.63 |
| 31 | 1.34 | 0.97 | 3.22 | 3.10 | 3.82 | 3.37 | 3.23 | 3.77 | 0.62 | 0.42 | 0.66 | 0.76 | 0.61 | 0.55 | 0.84 | 0.65 |
| 32 | 1.42 | 0.82 | 3.54 | 3.21 | 4.56 | 3.48 | 3.27 | 4.22 | 0.64 | 0.43 | 0.68 | 0.78 | 0.63 | 0.57 | 0.86 | 0.67 |
| 33 | 1.36 | 0.94 | 3.44 | 3.32 | 4.76 | 3.46 | 2.19 | 3.86 | 0.66 | 0.45 | 0.70 | 0.80 | 0.65 | 0.59 | 0.89 | 0.69 |
| 34 | 1.45 | 0.88 | 3.31 | 3.45 | 4.87 | 3.48 | 1.75 | 4.25 | 0.68 | 0.46 | 0.72 | 0.83 | 0.67 | 0.61 | 0.92 | 0.71 |
| 35 | | 0.89 | 3.07 | 3.92 | 5.09 | 3.59 | 1.89 | 4.11 | 0.70 | 0.47 | 0.74 | 0.85 | 0.69 | 0.63 | 0.95 | 0.73 |
| 36 | | 0.89 | 3.25 | 3.72 | 5.19 | 3.68 | 2.70 | 4.00 | 0.72 | 0.49 | 0.77 | 0.88 | 0.71 | 0.64 | 0.97 | 0.75 |
| 37 | 1.96 | 0.82 | 3.50 | 3.90 | 5.54 | 3.71 | 1.98 | 4.21 | 0.74 | 0.50 | 0.79 | 0.90 | 0.73 | 0.66 | 1.00 | 0.77 |
| 38 | | 0.82 | 3.31 | 2.96 | 5.50 | 3.65 | | 4.17 | 0.76 | 0.51 | 0.81 | 0.93 | 0.75 | 0.68 | | 0.79 |
| 39 | | 0.82 | 3.22 | 3.19 | 5.94 | 3.82 | | 4.13 | 0.78 | 0.53 | 0.83 | 0.95 | 0.76 | 0.70 | | 0.81 |
| 40 | | 0.82 | 3.19 | 3.54 | 5.93 | 3.71 | | 3.95 | 0.80 | 0.54 | 0.85 | 0.98 | 0.78 | 0.71 | | 0.83 |
| 41 | 1.74 | 0.77 | 3.07 | 3.06 | 5.95 | 3.91 | | 4.45 | 0.82 | 0.55 | 0.87 | 1.00 | 0.80 | 0.73 | | 0.85 |
| 42 | | 0.79 | 3.25 | | 5.99 | 4.06 | | 4.31 | 0.84 | 0.57 | 0.89 | | 0.82 | 0.75 | | 0.88 |
| 43 | | 0.72 | 3.42 | | 6.22 | 3.99 | | 4.67 | 0.86 | 0.58 | 0.91 | | 0.84 | 0.77 | | 0.90 |
| 44 | | 0.75 | 3.29 | | 6.73 | 4.32 | | 4.80 | 0.88 | 0.59 | 0.94 | | 0.86 | 0.79 | | 0.92 |
| 45 | | 0.74 | 3.16 | | 6.64 | 4.24 | | 4.59 | 0.90 | 0.61 | 0.96 | | 0.88 | 0.80 | | 0.94 |
| 46 | | 0.80 | 2.63 | | 6.00 | 4.33 | | 4.56 | 0.92 | 0.62 | 0.98 | | 0.90 | 0.82 | | 0.96 |
| 47 | | 0.83 | 2.53 | | 5.82 | 4.42 | | 5.31 | 0.94 | 0.64 | 1.00 | | 0.92 | 0.84 | | 0.98 |
| 48 | | 0.90 | | | 6.61 | 4.66 | | 3.27 | 0.96 | 0.65 | | | 0.94 | 0.86 | | 1.00 |
| 49 | | 0.88 | | | 6.16 | 4.61 | | | 0.98 | 0.66 | | | 0.96 | 0.88 | | |
| 50 | | 0.98 | | | 6.49 | 4.39 | | | 1.00 | 0.68 | | | 0.98 | 0.89 | | |
| 51 | | 0.95 | | | 5.49 | 4.29 | | | | 0.69 | | | 1.00 | 0.91 | | |
| 52 | | 0.83 | | | | 4.39 | | | | 0.70 | | | | 0.93 | | |

*Appendix 2—table 2 Continued on next page*

*Appendix 2—table 2 Continued*

| CA # | L-H ratios | | | | | | | | Relative distance (caudal #/total # of caudals) | | | | | | | |
| | Spino. | Mosa. | Ambly. | Alli. | Basil. | Hydro. | Trit. | Intel. | Spino. | Mosa. | Ambly. | Alli. | Basil. | Hydro. | Trit. | Intel. |
|---|---|---|---|---|---|---|---|---|---|---|---|---|---|---|---|---|
| 53 | | 0.91 | | | | 4.52 | | | | 0.72 | | | | 0.95 | | |
| 54 | | 0.91 | | | | 4.53 | | | | 0.73 | | | | 0.96 | | |
| 55 | | 0.88 | | | | 4.18 | | | | 0.74 | | | | 0.98 | | |
| 56 | | 0.96 | | | | 3.97 | | | | 0.76 | | | | 1.00 | | |
| 57 | | 0.89 | | | | | | | | 0.77 | | | | | | |
| 58 | | 1.04 | | | | | | | | 0.78 | | | | | | |
| 59 | | 0.96 | | | | | | | | 0.80 | | | | | | |
| 60 | | 0.94 | | | | | | | | 0.81 | | | | | | |
| 61 | | 1.08 | | | | | | | | 0.82 | | | | | | |
| 62 | | 0.99 | | | | | | | | 0.84 | | | | | | |
| 63 | | 0.86 | | | | | | | | 0.85 | | | | | | |
| 64 | | 0.88 | | | | | | | | 0.86 | | | | | | |
| 65 | | 1.06 | | | | | | | | 0.88 | | | | | | |
| 66 | | 1.03 | | | | | | | | 0.89 | | | | | | |
| 67 | | 0.96 | | | | | | | | 0.91 | | | | | | |
| 68 | | 1.02 | | | | | | | | 0.92 | | | | | | |
| 69 | | 1.01 | | | | | | | | 0.93 | | | | | | |
| 70 | | 0.99 | | | | | | | | 0.95 | | | | | | |
| 71 | | 1.00 | | | | | | | | 0.96 | | | | | | |
| 72 | | 1.01 | | | | | | | | 0.97 | | | | | | |
| 73 | | 1.01 | | | | | | | | 0.99 | | | | | | |
| 74 | | 0.98 | | | | | | | | 1.00 | | | | | | |

# Appendix 3

## Crocodylian foot paddle size and scaling

We measured forefoot, hind foot, and tail areas in five species of extant crocodylians by photographing appendages of individuals in captivity (*Appendix 3—table 1*).

**Appendix 3—table 1.** Forefoot, hind foot, and tail area and other data in five species of extant crocodylians.

American alligator (1–7, *Alligator mississippiensis*); Schneider's dwarf crocodile (8, *Paleosuchus trigonatus*); broad-snouted caiman (9, *Caiman latirostris*); spectacled caiman (10, *Caiman crocodilus*); African dwarf crocodile (11, *Osteolaemus tetraspis*).

| No. | Total length (cm) | Snout-vent length (cm) | Skull length (cm) | Post skull width (cm) | Tail length (cm) | Mass (kg) | Sex (m, f) | Age est. (year) | Hand (cm) | **Foot (cm)** | **Tail (cm)** |
|-----|-------|-------|------|------|-------|-------|---|----|------|-------|--------|
| 1 | 65.0 | 32.0 | 9.0 | 5.5 | 35.5 | 1.15 | f | 2 | 5.7 | 9.8 | 90.9 |
| 2 | 69.0 | 36.0 | 9.8 | 5.6 | 35.0 | 1.05 | f | 12 | 7.0 | 13.2 | 102.8 |
| 3 | 85.0 | 42.5 | 11.5 | 6.8 | 43.0 | 1.95 | f | 3 | 10.9 | 23.6 | 114.1 |
| 4 | 105.4 | 52.2 | 14.0 | 8.5 | 54.5 | 3.65 | f | 4 | 18.3 | 39.9 | 267.7 |
| 5 | 154.0 | 78.0 | 22.0 | 14.3 | 77.5 | 15.45 | m | 12 | 46.2 | 93.3 | 655.2 |
| 6 | 167.0 | 85.0 | 21.8 | 15.5 | 86.8 | 22.95 | m | ? | 51.4 | 96.0 | 717.9 |
| 7 | 211.5 | 104.5 | 25.3 | 18.5 | 108.0 | 39.10 | m | 23 | 69.5 | 146.5 | 1372.5 |
| 8 | 86.3 | 47.6 | 12.6 | 8.5 | 37.0 | 2.85 | f | 5 | 8.4 | 19.8 | 172.1 |
| 9 | 75.5 | 38.0 | 8.6 | 6.3 | 38.0 | 1.55 | f | 3 | 5.8 | 18.3 | 141.6 |
| 10 | 108.8 | 55.5 | 14.0 | 10.0 | 55.5 | 4.95 | f | 5 | 12.1 | 29.6 | 272.1 |
| 11 | 68.2 | 33.3 | 9.4 | 6.2 | 32.4 | 1.15 | f | 4 | 6.3 | 17.1 | 99.0 |

## Appendix 4

### Inland fossils referable to *Spinosaurus*

Inland fossils referable to *Spinosaurus* sp. were discovered in 1970 at the locality Gara Samani in the Béchar Basin of Algeria (*Appendix 4—table 1*). The most complete specimen is a snout (MNHN SAM 124) comparable in size to subadult *S. aegyptiacus* and eventually described as *S. maroccanus* (*Taquet and Russell, 1998*). We consider its specific status in doubt. *S. maroccanus* is based on isolated vertebrae from the Kem Kem Group in Morocco (*Russell, 1996*) and has recently been reduced to a junior synonym of *S. aegyptiacus* (*Ibrahim et al., 2020a*).

**Appendix 4—table 1.** Fossil material referable to *Spinosaurus* from inland basins in Algeria and Niger. D, dorsal.

| Specimens | Description |
| --- | --- |
| MNHN SAM 124 | Snout with preserved length of 62 cm (Gara Samani, Béchar Basin, Algeria) compared to ~1 m for adult snout MSNM V4047 |
| MNHN SAM 125 | Premaxilla fragment (Gara Samani, Béchar Basin, Algeria) |
| MNHN SAM 125–8 | Two cervical vertebra, one dorsal vertebra (Gara Samani, Béchar Basin, Algeria) |
| MNBH IGU11 | ~D1 short, very low oval centrum with low parapophysis, very strong ventral keel (In Abangharit, Iullumeden Basin, Niger) |
| MNBH EGA1 | Both maxillae and a portion of the alveolar edge of the right dentary with tooth roots within alveoli (Égaro North, Chad Basin, Niger) |
| MNBH EGA2 | Isolated tooth with root (Égaro North, Chad Basin, Niger) |

Inland fossils referable to *Spinosaurus* sp. come from two areas of outcrop of the Cenomanian-age Echkar Formation in Niger (*Appendix 4—table 1*). The first locality is near In Abangharit, which is northeast of Agadez and yielded isolated teeth and an anterior dorsal centrum initially referred to *Carcharodontosaurus iguidensis* (*Appendix 4—figure 1*). Reassigned to *Spinosaurus* sp., this vertebra is very close in form to the first dorsal centrum of *S. aegyptiacus* from Morocco (*Ibrahim et al., 2020a*: Fig. 128A–D) and Egypt (*Stromer, 1934*: pl. 1, Fig. 2), although differences suggest it may pertain to a distinct species. For the time being, reference is made only to the genus *Spinosaurus*.

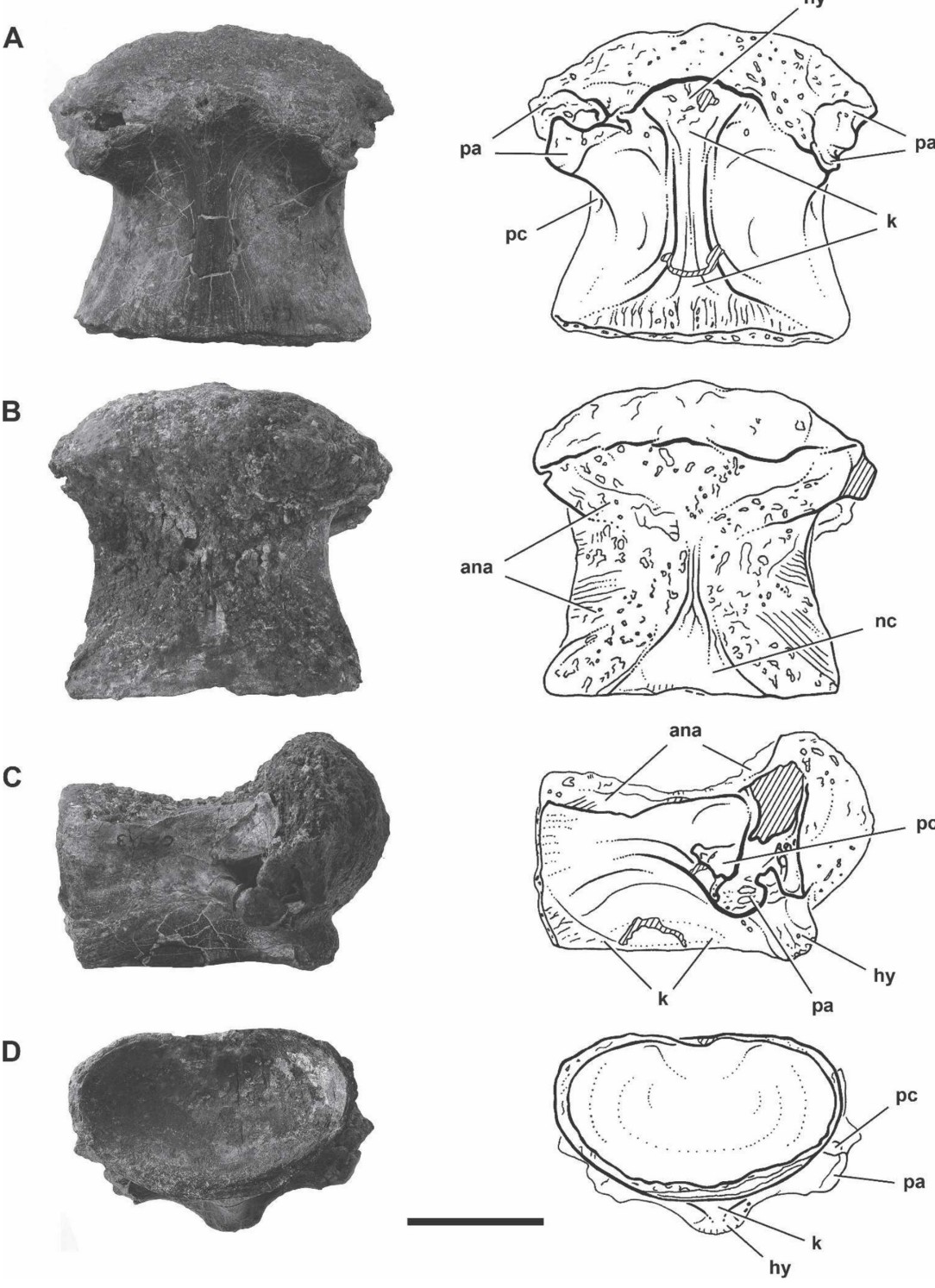

**Appendix 4—figure 1.** Anterior dorsal centrum referable to *Spinosaurus* sp. (from ***Brusatte and Sereno, 2007***: Fig. 9). Photographs and line drawings of MNBH IGU11 from the Echkar Formation (Cenomanian) of Niger in ventral (**A**) dorsal (**B**), right lateral (**C**), and posterior (**D**) views. Cross-hatching indicates broken bone. ana, articular surface for the neural arch; hy, hypapophysis; k, ventral keel; nc, neural canal; pa, parapophysis; pc, pleurocoel. Scale bar, 5 cm.

In 2019, a fragmentary snout preserving most of the right and left maxillae of *Spinosaurus* sp. was recovered from a second locality southeast of Agadez also in the Echkar Formation (***Figure 7B***). The material also includes a partial right dentary (MNBH EGA1) and subconical crowns and long

tapering roots (MNBH EGA2). The jaw bones, which closely resemble *S. aegyptiacus* although possibly a new species, are comparable in size to the subadult holotypic and neotypic skeletons, or about 75% of the size of the large snout recovered in Morocco (*Dal Sasso et al., 2005*). They were found in overbank deposits near the remains of rebbachisaurid and titanosaurian sauropods and evidence of a vertebrate fauna common to Cenomanian sites across northern Africa (including *Carcharodontosaurus*, lungfish tooth plates, sawfish rostral teeth, etc.).

## Appendix 5

### Phylogenetic analysis of Spinosauridae

Phylogenetic analysis of Spinosauroidea
We conducted a phylogenetic analysis within Spinosauroidea to better understand the distribution and evolution of putative display and semiaquatic features. The analysis includes 15 terminal taxa (9 spinosaurids) scored for 120 characters (49% cranial, 51% postcranial), 25 of which are newly introduced (marked NEW). Using the phylogenetic analysis program TNT (*Goloboff et al., 2008*), we used a heuristic search using 1000 Wagner tree replicates with 10 tree bisection reconnections per replicate. This generated a single most-parsimonious tree of 129 steps with consistency and retention indices of 0.806 and 0.854, respectively (*Appendix 5—figure 1*).

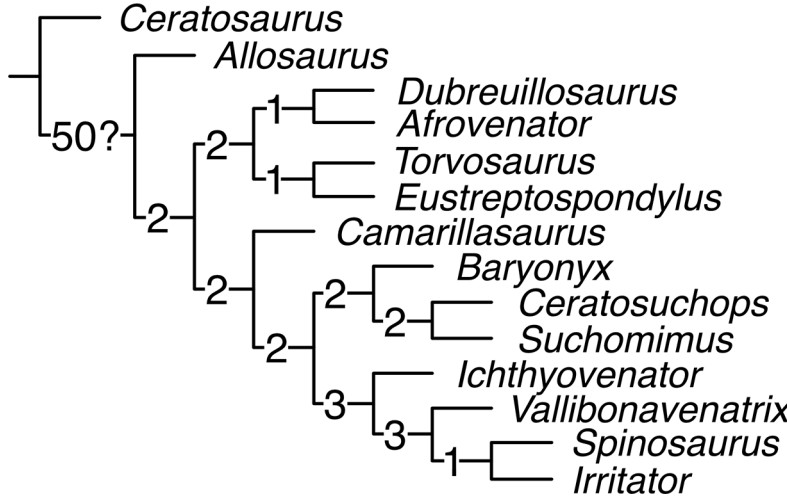

**Appendix 5—figure 1.** Phylogenetic tree for Spinosauroidea. Single most-parsimonious tree for 12 spinosauroid terminal taxa (9 spinosaurids) and 120 characters split evenly between the cranium (49%) and postcranium (51%), showing decay values (Consistency index = 0.81, Retention Index = 0.85).

*Camarillasaurus cirugedae*, originally described as a ceratosaur (*Sánchez-Hernández and Benton, 2012*), has been reinterpreted as a megalosauroid or possibly a spinosaurid (*Rauhut et al., 2019*; *Malafaia et al., 2020*; *Samathi et al., 2021*). Our analysis tentatively positions *Camarillasaurus* as the most basal spinosaurid, although additional remains of this fragmentary taxon are needed.

Recently, baryonychine material from the Wessex Formation on the Isle of Wight was described as two new genera, *Ceratosuchops inferodios* and *Riperovenator milnerae* (*Barker et al., 2021*). Both are close in form to *S. tenerensis* and differ from each other only in minor ways, with these disparities limited to overlapping bones (premaxillae, portions of the braincase). Some doubt remains regarding their association as single specimens as neither were found in association at single sites. The distinguishing features in the premaxillae (low narial tuberosity in one, not preserved in the other) and in the shape or depth of braincase fossae, the relative thickness of laminae and other minutiae of the braincase could well be due to individual variation (for *Allosaurus fragilis*, see *Chure and Madsen, 1996*). Several of these supposedly distinguishing features seem to occur on one or the other side of a well-preserved braincase of *S. tenerensis* (MNBH GAD43). The configuration of the orbital margin and form of the swollen postorbital brow in *S. tenerensis* (*Appendix 5—figure 2*) is similar to that in *Riparovenator* (it was cited as a distinguishing feature). We tentatively score the Wessex baryonychine material as a single taxon, *C. inferodios*, which resolves as the sister taxon to *S. tenerensis* (*Appendix 5—figure 1*).

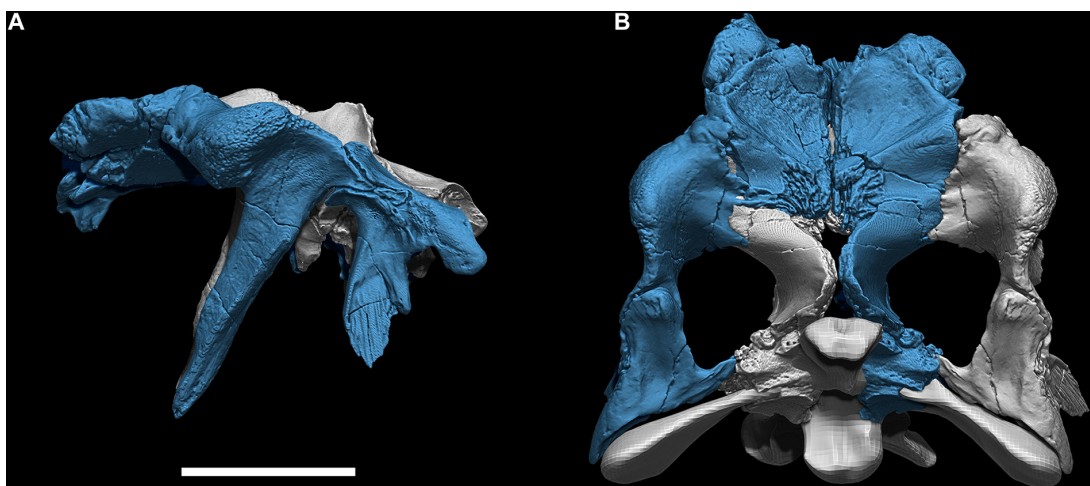

**Appendix 5—figure 2.** Posterior skull roof of the baryonychine spinosaurid *Suchomimus tenerensis* (field no. GAD302). Composite restoration of the posterior skull roof of *Suchomimus tenerensis* in (**A**) lateral and (**B**) dorsal views showing a swollen postorbital brow and narrow orbital notch limiting the frontal orbital margin. Based on original bone (blue), reflected original bone (gray), and reconstructed paroccipital processes.

With more spinosaurid remains coming to light in the decades before 2000, *Sereno et al., 1998* split the group into sister clades, Baryonychinae and Spinosaurinae. Nearly all subsequent analyses including the current effort resolve this basal split (*Allain et al., 2012*; *Carrano et al., 2012*; *Arden et al., 2019*; *Malafaia et al., 2020*; *Barker et al., 2021*), with one outlier showing *Baryonyx*, *Suchomimus*, and Spinosaurinae as an unresolved polytomy (*Sales and Schultz, 2017*). Larger-scale analyses that incorporate theropods farther afield have not added any resolution or insights to an understanding of spinosauroid interrelationships. Although we use some of the relevant characters in those analyses (e.g., *Carrano et al., 2012*).

The holotypic specimen of *Irritator* shares important cranial apomorphies with *Spinosaurus* (an oval quadrate head, fewer and more spaced maxillary teeth, straight tooth crowns). Spinosaurid postcranial remains from the same formation yielding *Irritator*, in addition, share additional apomorphies with *Spinosaurus* (posterior dorsal neural spines anteroposteriorly shorter than centrum length, a proportionately lower iliac blade, and a ventrally directed iliac pubic peduncle). *Ichthyovenator* (*Allain et al., 2012*) and the recently published *Vallibonavenatrix* (*Malafaia et al., 2020*) are also recovered as spinosaurines on the basis of a few unambiguous postcranial synapomorphies (see below).

## Character list
### Skull

1. Snout (preorbital region of skull), length relative to antorbital fenestra length: less (0) or more (1) than three times. (*Sereno et al., 1998*, char. 7)
2. Jaws (premaxillae and dentary), anterior end, form: convergent (0); expanded into a premaxillary/dentary rosette (1). (*Sereno et al., 1998*, char. 6)
3. Premaxilla, form of premaxilla-nasal suture: V-shaped (0); W-shaped (1). (*Carrano et al., 2012*, char. 5)
4. Premaxilla, external nares, proportions and position: shorter than premaxilla ventral to nares, angle between anterior and alveolar margins >75° (0); longer than body ventral to nares (1). (*Carrano et al., 2012*, char. 6)
5. Premaxillae, inter-premaxillary suture at maturity, form: open (0); fused (1). (*Sereno et al., 1998*, char. 10)
6. Premaxilla-maxilla articulation, form: scarf or butt joint (0); interlocking (1). (*Sereno et al., 1998*, char. 11)
7. Premaxilla-maxilla suture, lateral surface, subnarial foramen, shape: foramen (0); dorsoventrally directed channel (1). (*Carrano et al., 2012*, char. 10)

8. Premaxilla, ventral margin, shape in lateral view: straight to convex (0); concave (1). (*Cau, 2018*, char. 1485)

9. Premaxilla, lateral and dorsal surface, extensive pitting of neurovascular foramina: absent (0); present (1). (*Barker et al., 2021*, char. 1796)

10. Maxilla, anterior ramus, length relative to maximum depth: 70% (0); 100% or more (1). (*Sereno et al., 1998*, char. 1)

11. Maxilla, contribution to the narial fossa: no contribution (0); contributes partially or totally (1). (Modified from *Longrich and Currie, 2009*, char. 7)

12. Maxilla, antorbital fossa, width of ventral margin relative to depth of posterior ramus of maxilla: more (0) or less (1) than 30%. (*Sereno et al., 1998*, char. 40)

13. Maxilla, antorbital fossa, anterior margin: rounded (0); squared (1). (*Carrano et al., 2012*, char. 23)

14. Maxilla, subcircular depression in the anterior corner of the antorbital fossa: absent (0); present (1). (*Sereno et al., 1998*, char. 41)

15. Maxilla, anteromedial process, shape: fluted prong (0); plate (1). (*Sereno et al., 1998*, char. 12)

16. Maxilla, anteromedial process, anterior extension: as far as (0) or far anterior to (1) the anterior margin of the maxilla. (*Sereno et al., 1998*, char. 13)

17. Maxilla, antorbital fossa, size relative to orbit: larger (0); smaller (1). (*Sereno et al., 1998*, char. 9)

18. Maxilla, anteroventral margin, curvature: not curved (0); dorsomedially curved (1). (Modified from *Cau, 2018*, char. 731; *Tykoski, 2005*)

19. Nasal, posterior process, overlap of frontal in articulation: absent or limited (0); extensive, in particular medially, on almost or all the process dorsal surface (1). (Modified from *Cau, 2018*, char. 1500)

20. Jugal, posterior ramus, depth relative to orbital ramus: less (0); more (1). (*Sereno et al., 1998*, char. 43)

21. Lacrimal, anterior and ventral rami, angle of divergence: 75° to 90° (0); 30° to 45° (1). (*Sereno et al., 1998*, char. 15)

22. Lacrimal foramen, position relative to ventral process: near the base (0); at mid-height (1). (*Sereno et al., 1998*, char. 42)

23. Lacrimal, anterior ramus, length relative to ventral ramus: more (0) or less (1) than 65%. (*Sereno et al., 1998*, char. 2)

24. Postorbital, ventral process, cross section of distal half: subcircular (0); U-shaped (1). (*Sereno et al., 1998*, char. 44)

25. Postorbital, supraorbital shelf (boss) formed mostly by palpebral: absent (0); present (1). (Modified from *Carrano et al., 2012*, char. 61)

26. Frontal, postorbital facet, anterior depth: less than 2/5 facet length (0); more than 2/5 facet length (1). (*Barker et al., 2021*, *Source data 2*)

27. Frontal, shape of the lateral margin in dorsal view: describes a smooth transition between the anterior half and the postorbital process (0); an abrupt transition between the anterior half and the postorbital process (1). (*Senter et al., 2010*, char. 44)

28. Prefrontal, boss-like process: absent (0); present (1). (*Barker et al., 2021*, *Source data 2*)

29. Parietal, length: less than 3/4 of the frontal (0); subequal or more than 3/4 of the frontal (1). (*Cau, 2018*, char. 78)

30. Quadrate, head, shape: oval (0); subquadrate (1). (*Sereno et al., 1998*, char. 27)

31. Quadrate, foramen, size: foramen (0); broad fenestra (1). (*Sereno et al., 1998*, char. 28)

32. Quadrate, medial foramina adjacent to condyles: absent (0); present (1). (*Carrano et al., 2012*, char. 84)

33. Quadrate, foramen margin, placement: at mid-height or dorsal (0); ventral, close to mandibular condyles (1). (Modified from *Loewen et al., 2013*, char. 158)

34. Basisphenoid, width of the interbasipterygoidal web: thin; 40% of occipital condyle width or less (0); thick, more than 40% occipital condyle width (1). (Modified from *Barker et al., 2021*, *Source data 1*)

35. Basisphenoid, basipterygoid process, exposure of the ventral surface in lateral view: thick (0); reduced (1). (*Barker et al., 2021*, *Source data 2*)

36. Basisphenoid, basipterygoid process, shape of lateral margin in ventral view: flat or slightly concave (0); convex (1). (*Barker et al., 2021*, *Source data 2*)

37. Basioccipital, position of subcondylar recess: dorsoventrally tall, recess reaches the occipital condyle neck (0); ventrally restricted, surface directly below condyle convex (1). (*Barker et al., 2021*, *Source data 2*)
38. Basioccipital, width of subcondylar recess relative to occipital condyle width: narrow, 0.5 times or less (0); wide, greater than 0.5 times (1). (Modified from *Barker et al., 2021*, *Source data 2*)
39. Basioccipital, thick crests bordering subcondylar recess laterally: present (0); absent (1). (*Barker et al., 2021*, *Source data 2*)
40. Basioccipital, contribution to foramen magnum: large, exoccipitals widely separated (0); reduced, exoccipitals closely placed (1). (*Barker et al., 2021*, *Source data 2*)
41. Basioccipital, proportions relative to basisphenoid (measured along midline ventral to occipital condyle to interbasipterygoidal web), in posterior view: shorter (0); longer (1). (*Barker et al., 2021*, *Source data 2*)
42. Otoccipitals, angle of projection of paroccipital processes: posterolaterally (0); laterally or subhorizontal (1). (*Barker et al., 2021*, *Source data 2*)
43. Splenial foramen, size: small (0); large (1). (*Sereno et al., 1998*, char. 16)
44. Dentary, shape of anterior end in lateral view: blunt and unexpanded (0); dorsoventrally expanded, rounded and slightly upturned (1); squared off in lateral view via anteroventral process (2). (*Carrano et al., 2012*, char. 120)
45. Premaxilla and anterior dentary, interdental septa spacing: regular (0); alternate (alveoli result paired) (1). (Modified from *Cau, 2018*, char. 1614)

## Dental

46. Teeth, distal, curvature: present, marked (0); reduced or non-curved (1). (*Sereno et al., 1998*, char. 35; modified after *Hendrickx et al., 2019*)
47. Teeth, maxillary and dentary, serrations: present (0); absent (1). (*Sereno et al., 1998*, char. 17)
48. Teeth, distal, midcrown cross-section: elliptical (0); circular (1). (*Sereno et al., 1998*, char. 36; modified after *Hendrickx et al., 2019*)
49. Teeth, distal, crown striations (flutes/apicobasal ridges): absent (0); present (1). (*Sereno et al., 1998*, char. 18; modified after *Hendrickx et al., 2019*)
50. Teeth, enamel ornamentation: absent (0); present (1). (Modified from *Carrano et al., 2012*, char. 143)
51. Teeth, enamel ornamentation type: extending as bands across labial and lingual tooth surfaces (0); pronounced marginal enamel wrinkles (1); pronounced deeply veined/anastomous texture (2). (Modified from *Carrano et al., 2012*, char. 143)
52. Teeth, basalmost root shape: broad (0); strongly tapered (1). (*Sereno et al., 1998*, char. 21)
53. Premaxilla, tooth count: 3–4 (0); 6–7 (1). (*Sereno et al., 1998*, char. 19)
54. Premaxillary tooth 1, size: slightly smaller (0) or much smaller (1) than crowns 2 and 3. (*Sereno et al., 1998*, char. 38)
55. Premaxilla, diastemata within the premaxillary rosette: narrow (0); broad (1). (*Sereno et al., 1998*, char. 39)
56. Maxillary teeth, mid-tooth spacing: adjacent (0); with intervening space/diastemata (1). (*Sereno et al., 1998*, char. 20)
57. Dentary, tooth count: up to 15 (0); ≥15 (1). (*Sereno et al., 1998*, char. 26)
58. Dentary teeth, spacing: adjacent (0); with intervening space (1). (*Sereno et al., 1998*, char. 37)
59. Paradental laminae: present (0); absent (1). (*Sereno et al., 1998*, char. 14)

## Axial

60. Cervical vertebrae, middle, shape of anterior pneumatic foramina: round (0); anteroposteriorly elongate (1). (*Carrano et al., 2012*, char. 169)
61. Cervical vertebrae, posteriormost, ventral keel: absent or developed as a weak ridge (0); pronounced, around 1/3 the height of centrum and inset from lateral surfaces (1). NEW
62. Cervical vertebrae, prezygapophyseal facets, elongation related to width: as long as wide (0); longer than wide (1); wider than longer (2). NEW

63. Cervical vertebrae, middle and posterior centra width relative to centra height: taller than wide or round (0); wider than tall (1). NEW

64. Cervical vertebrae, anterior post-axial neural spines in lateral view: longer than tall (0); taller than long (1). (*Cau, 2018*, char. 212)

65. Dorsal vertebrae, anterior centra, depth of ventral keel relative to total centrum height: absent or less than ¼ (0); blade-shaped, more than ¼ (1). (*Sereno et al., 1998*, char. 22)

66. Dorsal vertebrae, anterior centra, ventral processes anterior to the keel (hypapophysis): poorly developed (0); strongly developed (1). (Modified by *Cau, 2018*, char. 225; from *Rauhut, 2003*)

67. Dorsal vertebrae, middle, centra length relative to height: 1.4 < times centrum height (0); ≥1.4 times centrum height (1). NEW

68. Dorsal vertebrae, anterior parapophyses, size: less (0) or more (1) than half-depth of anterior facet of centrum. (*Cau, 2018*, char. 1740)

69. Dorsal vertebrae, anterior centra, pneumatic foramen, size relative to parapophysis: larger or equal (0); smaller (1). NEW

70. Dorsal vertebrae, anterior, prezygapophyseal facets, elongation related to width: as long as wide (0); longer than wide (1); wider than longer (2). NEW

71. Dorsal vertebrae, mid-posterior, excavated prezygo-para-diapophyseal fossa (prpadf) delimited by the paradiapophyseal lamina (ppdl): absent (0); present (1). NEW (after *Malafaia et al., 2020*)

72. Dorsal vertebrae, middle and posterior centra, pneumatic foramina: absent (0); present (1). (Modified from *Cau, 2018*, char. 1350)

73. Dorsal vertebrae, height of neural spines relative to centrum height: low,<1.3× (0); moderate, 1.3–1.8× (1); tall, <1.8× (2). (Modified from *Carrano et al., 2012*, char. 193)

74. Dorsal vertebrae, posterior neural spines, basal webbing: absent (0); present (1). (*Sereno et al., 1998*, char. 24)

75. Dorsal vertebrae, posterior neural spines, accessory centrodiapophyseal lamina: absent (0); present (1). (*Sereno et al., 1998*, char. 25)

76. Dorsal vertebrae, middle and posterior, neural spine anteroposterior length at base relative to centrum length: subequal (0); shorter (1). NEW

77. Dorsal vertebrae, middle-posterior parapophyses, development: distinct elevated process (0); reduced, knob-shaped (1). NEW

78. Dorsal vertebrae, accessory centrodiapophyseal lamina: absent (0); present (1). (Modified from *Benson et al., 2010*, char. 2015)

79. Sacrum, centrum pneumaticity: absent (0); present (1). (Modified from *Carrano et al., 2012*, char. 196)

80. Sacrum, centrum pneumaticity type: pleurocoelous fossae (0); pneumatic foramina (1). (Modified from *Carrano et al., 2012*, char. 196)

81. Sacrum, neural spines: without distal anteroposterior expansion (0); with distal expansion contacting adjacent spines (1). NEW

82. Caudal vertebrae, anterior, morphology of ventral surface: flat (0); groove (1); ridge (2). (*Carrano et al., 2012*, char. 202)

83. Caudal vertebrae, anterior, well-marked spino-diapophyseal lamina: absent (0); present (1). NEW

84. Caudal vertebrae, anterior neural arches, ventral rib laminae: absent (0); present (1). (*Cau, 2018*, char. 358)

85. Caudal vertebrae, anterior neural arches, anterolateral surface, deep triangular prezygocostal fossa delimited by two laminae: absent (0); present (1). (*Cau, 2018*, char. 1605; modified from *Brusatte et al., 2010*)

86. Caudal vertebrae, anterior neural arches, hyposphene: absent (0); present (1). (*Cau, 2018*, char. 359)

87. Caudal vertebrae, position of transition point (caudofemoralis and ilio-ischiocaudalis mm. correlates): at or beyond CA20 (0); around CA15-19 (1). NEW

88. Caudal vertebrae, middle, height of neural spines relative to centrum height: low, ≥1.3× (0); moderate, 1.3–2.0× (1); tall, 2.0–4.0× (2); extreme elongation >4× (3). NEW

89. Caudal vertebrae, distal, height of neural spines relative to centrum height: low, ≥1.3× (0); moderate, 1.3–2.0× (1); tall, 2.0–4.0× (2); extreme elongation >4× (3). NEW

90. Caudal vertebrae, middle, morphology of neural spines: rod-like and posteriorly inclined (0); subrectangular and sheet-like (1); rod-like and vertical (2). (*Carrano et al., 2012*, char. 207)
91. Caudal vertebrae, middle, centrum elongation relative to centrum height: elongated >1.6× (0); not elongated ≤1.6× (1). NEW
92. Caudal vertebrae, middle and posterior, prezygapophyses: elongated, projected beyond the anterior rim of the centrum (0); shortened, barely reaching the anterior rim of the centrum (1). NEW (after observations in *Samathi et al., 2021*)
93. Chevrons, anterior and posterior longitudinal groove: absent (0); present (1). NEW
94. Chevrons, anterior and posterior surfaces: without distinctive features (0); with longitudinal groove widened as a fossa (1). NEW
95. Chevrons, posterior elongation compared with anterior and middle ones: shorter (0); as elongated (1). NEW
96. Chevrons, anterior process: absent (0); present (1). (*Carrano et al., 2012*, char. 217)

## Appendicular

97. Coracoid, posterior process, shape: low and rounded (0); crescentic (1). (*Sereno et al., 1998*, char. 29)
98. Appendicular bones, marrow cavity: present (0); reduced to barely present (1). NEW
99. Humerus, deltopectoral crest, length relative to humeral length: less (0) or more (1) than 45%. (*Sereno et al., 1998*, char. 3)
100. Humerus, deltopectoral crest, orientation of apex: anterior (0), lateral (1). (*Sereno et al., 1998*, char. 31)
101. Humerus, trochanters, size: low and rounded (0); hypertrophied and pointy (1). (*Sereno et al., 1998*, char. 30)
102. Humerus, internal tuberosity, size: low and rounded (0); hypertrophied (1). (*Sereno et al., 1998*, char. 32)
103. Radius (forearm), length relative to humeral length: more (0) or less (1) than 50%. (*Sereno et al., 1998*, char. 4)
104. Radius, external tuberosity and ulnar internal tuberosity, size: low and rounded (0); hypertrophied (1). (*Sereno et al., 1998*, char. 33)
105. Manual digit I–ungual, length relative to the depth of proximal end: 2.5 (0) or 3 (1) times. (*Sereno et al., 1998*, char. 5)
106. Ilium, ventrolateral development of supraacetabular crest: large/pendant 'hood' (0); reduced shelf (1). (*Carrano et al., 2012*, char. 267)
107. Ilium, lateral vertical crest dorsal to the acetabulum: absent (0); present (1). (*Rauhut, 2003*, char. 6)
108. Ilium, shape of posterior margin of postacetabular process: convex (0); concave (1); straight (2); with prominent posterodorsal process but lacking posteroventral process (3). (*Carrano et al., 2012*, char. 280)
109. Ilium, orientation of pubic peduncle: mostly ventral (0); mostly anterior or 'kinked' double facet with anterior and ventral components (1). (*Carrano et al., 2012*, char. 268)
110. Ilium, pubic peduncle length to width ratio: ≤1 (0); 1–2 (1); >2 (2). (Modified from *Carrano et al., 2012*, char. 272)
111. Ilium, brevis fossa, lateral and medial margins, orientation in ventral view and development of fossa: subparallel, narrow fossa (0); posteriorly diverging, expanded fossa (1). (Modified by *Cau, 2018*, char. 592; *Holtz, 2000*; *Rauhut, 2003*)
112. Puboischiadic plate, foramina/notches: closed along midline, 3 fenestrae (0); open along midline, 1 fenestra (obturator foramen of pubis) and 1–2 notches (1); open along midline, 0 fenestrae, 1–2 notches (2). (*Carrano et al., 2012*, char. 281)
113. Pubis, distal pubic foot, size: moderate to large (0); reduced to a small flange (1). (Modified from *Sereno et al., 1998*, char. 34)
114. Pubis, length relative to ischium: longer (0); subequal or shorter (1). NEW
115. Ischium, distal half, cross section: laminar, strongly mediolaterally compressed (0); robust, rod-like (1). (*Cau, 2018*, char. 425)

116. Femur, fourth trochanter position of distalmost end: proximal most 1/3 (0); almost at the half of the femur (1). NEW

117. Femur, oblique ligament groove on posterior surface of head: shallow, groove bounding lip does not extend past posterior surface of head (0); deep, bound medially by well-developed posterior lip (1). (*Carrano et al., 2012*, char. 304)

118. Tibia and/or femur, length compared to posterior dorsal centra length: more (0) or less (1) than five times. (*Cau, 2018*, char. 245)

119. Tibia, proximal diaphysis, length-width ratio: smaller than 2 (0); greater than 2 (1). NEW (after *Samathi et al., 2021*)

120. Pedal ungual phalanges, ventral side: concave (0); flat (1). NEW

## Character-taxon matrix

See *Source data 1* and *Source data 2*.

## Apomorphy list

## Spinosauridae (with *Camarillasaurus*)

Ch. 48: 0 →1 Maxillary lateral teeth with circular midcrown cross section.
Ch. 92: 0 →1 Middle and posterior caudal vertebrae with short prezygapophyses.
Ch. 119: 0 →1 Tibia with a proximal diaphysis with length + width ratio greater than 2.

## Spinosauridae (minus *Camarillasaurus*)

Ch. 49: 0 →1 Maxillary lateral teeth with fluting (apico-basal ridges).
Ch. 94: 0 → 1 Anterior chevrons with longitudinal groove widened as a fossa.

## Baryonychinae

Ch. 3: 0 → 1 V-shaped premaxilla-nasal suture.
Ch. 30: 0 →1 Square-shaped quadrate head.
Ch. 57: 0 →1 30 dentary teeth.
Ch. 64: 1 →0 Middle cervical vertebrae with neural spines longer than tall.
Ch. 74: 0 →1 Dorsal vertebrae, posterior neural spines with basal webbing.
Ch. 78: 0 →1 Dorsal vertebrae with accessory centrodiapophyseal lamina.
Ch. 97: 0 →1 Coracoid with crescentic posterior process.

## Ceratosuchopsini (*Suchomimus* + *Ceratosuchops*)

Ch. 26: 0 →1 Frontal, postorbital facet depth more than 2/5 facet length.
Ch. 27: 0→1 Frontal with abrupt transition between the anterior half and the postorbital process.
Ch. 40: 0 →1 Basioccipital contribution to foramen magnum reduced, with exoccipitals closely placed.

## Spinosaurinae

Ch. 63: 0 →1 Middle and posterior cervical vertebrae with centra wider than tall.
Ch. 66: 0 →1 Anterior dorsal vertebrae with strongly developed hypapophysis.
Ch. 69: 0→1 Anterior dorsal vertebrae with pneumatic foramen smaller than parapophysis.
Ch. 71: 0 →1 Middle and posterior dorsal vertebrae with deeply excavated prezygopara-diapophyseal fossa (prpadf).
Ch. 76: 0 →1 Posterior dorsal vertebrae with neural spine base shorter than centrum.
Ch. 77: 0→1 Middle and posterior dorsal vertebrae with reduced, knob-like parapophyses.
Ch. 84: 0 →1 Anterior caudal vertebrae with centro-costal laminae.
Ch. 85: 0 →1 Anterior caudal vertebrae with prezygo-costal fossa delimited by two laminae.

## *Vallibonavenatrix* **+ (***Spinosaurus + Irritator***)**

Ch. 67: 0 →1 Mid-dorsal vertebrae centra longer than 1.4 times centrum height.
Ch. 72: 0 →1 Middle and posterior dorsal centra with large pneumatic foramina.
Ch. 83: 0 →1 Anterior caudal vertebrae with well-marked spino-diapophyseal lamina.
Ch. 91: 0 →1 Shorter middle caudal centra.

## *Spinosaurus + Irritator*

Ch. 109: 0 →1 Ilium, orientation of pubic peduncle mostly anterior or 'kinked' double facet with anterior and ventral components.

