## [Editor Report]

This article evaluates the hypothesis that Spinosaurus was a specialized aquatic dinosaur, by developing a CT-based skeletal restoration and examining its hydrodynamic properties. In this reappraisal of the "aquatic hypothesis", new results support the alternative "semi-aquatic hypothesis". This article will be of interest to vertebrate paleontologists and functional morphologists, as well as wider academic and non-academic audiences.

---

## [Decision Letter]

**Decision letter after peer review:**

Thank you for submitting your article "Spinosaurus is not an aquatic dinosaur" for consideration by *eLife*. Your article has been reviewed by three peer reviewers, and the evaluation has been overseen by Min Zhu as Reviewer #1 and Reviewing Editor and Christian Rutz as the Senior Editor. The following individuals involved in the review of your submission have agreed to reveal their identity: Thomas R. Holtz (Reviewer #2); David Hone (Reviewer #3).

The reviewers have discussed their reviews with one another, and the Reviewing Editor has drafted this decision letter to help you prepare a revised submission.

Essential revisions:

The reviewers found the analyses presented in this article valuable, but have raised a range of questions that need to be carefully addressed in a revision; the full reports are appended below. Amongst other things, the state of the companion paper by Myhrvold et al. (in press) should be clarified, as well as its relationship to the present submission.

*Reviewer #1 (Recommendations for the authors):*

My major concern is the state of one paper (Myhrvold et al., in press), which provides critical data to support the key claims of this manuscript. As I can only find the preprint version of Myhrvold et al. (in press) on the web, the state of this relevant paper should be clarified.

The arguments in the section 'Paleohabitats and evolution' are not so convincing as those in the other sections. For instance, the significance of the inland location of the Spinosaurus fossils is not so crucial for the evaluation of the two hypotheses.

Line 244: "The large body size of *S. aegyptiacus* and antecedent species such as *S. tenerensis*" is misleading. There exist many forms intermediate between the two 'S.'. These two S. belong to the two genera.

Line 253: A definition of Stage 1 should be provided. "The earliest spinosaurids" is ambiguous. In Figure 8, Stage 1 and Stage 2 are represented by two nodes. Correct?

The conclusion section seems rather trivial. The contents should be summarized to present the major claims of this manuscript.

Lines 24-26: This sentence should be clarified. Delete "living reptiles with"?

Line 27: Do you mean "the evolution of spinosaurids" here?

Lines 101-103: semi-aquatic?

Line 163: To be clarified.

Line 201: The co-occurrence of *S. aegyptiacus* and *S. tenerensis* here is misleading, as the first S. represents Spinosaurus and the second Suchomimus. How about Su. tenerensis? The same for the rest.

Figure 1F: No cross.

Lines 242-255: Line numbers in wrong positions.

Line 273: Bottom of Figure 3A?

*Reviewer #2 (Recommendations for the authors):*

There are relatively few changes to recommend to this manuscript. The primary new data (the reconstruction) is well-documented and the justification for the decisions in its development is given.

Most importantly, I would be interested in seeing the inclusion of paddle-derived swimmer proportions in a revision. This is not because I consider Spinosaurus as a likely subaqueous flier; it is simply to remove arguments that might arise from the statement that appendage surface area is necessarily reduced in all secondarily-aquatic tetrapods. The existence of underwater fliers and other paddle-based swimmers suggests that not all marine tetrapods have reduced limbs. One might expect a higher appendage surface area/total body area value in marine turtles, penguins, plesiosaurs, and Megaptera (for instance) than seen in the marine taxa in Figure 5 here, and it might be valuable to include these data. That said, nothing in the anatomy of spinosaurid dinosaurs suggests them to have limb-based propulsion like subaqueous fliers, and indeed the authors here show the inadequacy of the propulsive ability of Spinosaurus feet.

*Reviewer #3 (Recommendations for the authors):*

While I have a number of issues with the manuscript as it stands that need more detail in the writing and citations of various sources, especially some further explanations of the methods in places, it is fundamentally sound. Additional details, some revision to the order in which information is presented and clarity over definitions is the main thing required to get this to a publishable standard. In short, this is very good and I think it's an excellent test of the aquatic hypothesis but it really needs a few points addressed.

1) There are various places where it's not very clear at all what you did methodologically. I'm no expert in deep biomechanics and can't follow all the detail anyway, but there are some very loose 'and then X was estimated' type sentences that really need a line about how or why, or a justification.

2) Some of the structure is off. There are sections in the Results which go through all the maths of a calculation which should surely be in the Methods. Similarly, there is no apparent Discussion section and this really blurs the line between results and discussion and especially when I think some methods are in there.

3) It really needs more clarity around details in some places. First of all, there are times when you should be citing some of the other literature on Spinosaurus and its relatives but also be more clear on definitions. One criticism of the aquatic hypothesis in its various guises has been a lack of clarity over what is actually meant by e.g. 'subaqueous foraging'. Your section on aquatic vs. semi-aquatic partially clears that up but both there and at the end of the paper it's not clear what you think Spinosaurus could or did do. For example, you say it could not dive, but don't say if you think it could swim at all (or at least better than other theropods) which would make this much stronger and prevent the perpetuation of uncertainty or confusion over what your results actually show.

---

## [Author Response]

Essential revisions:The reviewers found the analyses presented in this article valuable, but have raised a range of questions that need to be carefully addressed in a revision; the full reports are appended below. Amongst other things, the state of the companion paper by Myhrvold et al. (in press) should be clarified, as well as its relationship to the present submission.

Myhrvold et al., in press” cites a draft Matters Arising reply to Fabbri et al. (2022) in the journal Nature that was ultimately declined as a published exchange by that journal after its preprint posting. We are reworking and extending that effort now as a longer research paper, which has not been submitted. Nothing in the current resubmission depends on the conclusions of our ongoing research on shortcomings in methods used to measure bone compactness. Although we have removed reference to the preprint in the revision, we are also good with retention of that preprint citation, should that be the journal’s preference, as we stand behind the results and data presented in that short piece. But, again, the measure of compactness in hind limb bones does not materially affect the basis for any argument presented in the paper.

Reviewer #1 (Recommendations for the authors):The arguments in the section 'Paleohabitats and evolution' are not so convincing as those in the other sections. For instance, the significance of the inland location of the Spinosaurus fossils is not so crucial for the evaluation of the two hypotheses.

We feel these arguments are as convincing as any others, simply because there is no living or extinct exception to the geographic habitat limitation of large-bodied, fully aquatic vertebrates —to the marine realm rather than inland locations.

The conclusion section seems rather trivial. The contents should be summarized to present the major claims of this manuscript.

Concise listing of the major claims in the manuscript is what we aim to do with the numbered conclusions section of the manuscript, which distills sometimes involved arguments in a manuscript that has already exceeded suggested length.

Line 201: The co-occurrence of *S. aegyptiacus* and *S. tenerensis* here is misleading, as the first S. represents Spinosaurus and the second Suchomimus. How about Su. tenerensis? The same for the rest.

This is a common occurrence in the taxonomic literature (identical generic abbreviations); the two in question are clearly separated/identified by their species. Mosquito taxonomists, fairly uniquely, use a 2-letter abbreviation for genera. No one, to our knowledge, uses both a single and 2-letter abbreviation in the same paper.

Reviewer #2 (Recommendations for the authors):Most importantly, I would be interested in seeing the inclusion of paddle-derived swimmer proportions in a revision. This is not because I consider Spinosaurus as a likely subaqueous flier; it is simply to remove arguments that might arise from the statement that appendage surface area is necessarily reduced in all secondarily-aquatic tetrapods. The existence of underwater fliers and other paddle-based swimmers suggests that not all marine tetrapods have reduced limbs. One might expect a higher appendage surface area/total body area value in marine turtles, penguins, plesiosaurs, and Megaptera (for instance) than seen in the marine taxa in Figure 5 here, and it might be valuable to include these data. That said, nothing in the anatomy of spinosaurid dinosaurs suggests them to have limb-based propulsion like subaqueous fliers, and indeed the authors here show the inadequacy of the propulsive ability of Spinosaurus feet.

We implemented the major and only suggestion, adding vertebrate underwater fliers to the plot in Figure 5.

Reviewer #3 (Recommendations for the authors):2) Some of the structure is off. There are sections in the Results which go through all the maths of a calculation which should surely be in the Methods. Similarly, there is no apparent Discussion section and this really blurs the line between results and discussion and especially when I think some methods are in there.

We put in Methods all elaborate calculations and formulations. We feel that the very few formulae we included in functional discussions of a series of performance measures (speed, maneuverability, etc.) help readers understand our reasoning. We prefer not to move text to Methods simply because it includes a mathematical formula.